# Assembly of a unique membrane complex in type VI secretion systems of Bacteroidota

Thibault R. Bongiovanni[1,9], Casey J. Latario [2], Youn Le Cras [3], Evan Trus[2], Sophie Robitaille[2], Kerry Swartz[2], Danica Schmidtke[2,4], Maxence Vincent[1,9], Artemis Kosta [5], Jan Orth[6,7], Florian Stengel [6,7], Riccardo Pellarin[8], Eduardo P. C. Rocha [3], Benjamin D. Ross[2,4] ✉ & Eric Durand[1,9] ✉

The type VI secretion system (T6SS) of Gram-negative bacteria inhibits competitor cells through contact-dependent translocation of toxic effector proteins. In Proteobacteria, the T6SS is anchored to the cell envelope through a megadalton-sized membrane complex (MC). However, the genomes of Bacteroidota with T6SSs appear to lack genes encoding homologs of canonical MC components. Here, we identify five genes in *Bacteroides fragilis* (*tssNQOPR*) that are essential for T6SS function and encode a Bacteroidota-specific MC. We purify this complex, reveal its dimensions using electron microscopy, and identify a protein-protein interaction network underlying the assembly of the MC including the stoichiometry of the five TssNQOPR components. Protein TssN mediates the connection between the Bacteroidota MC and the conserved baseplate. Although MC gene content and organization varies across the phylum Bacteroidota, no MC homologs are detected outside of T6SS loci, suggesting ancient co-option and functional convergence with the non-homologous MC of Pseudomonadota.

The human gut is inhabited by a diverse and abundant bacterial community which contributes to many aspects of host physiology[1]. In dense environments like the large intestine, approaching an estimated $10^{11}$-$10^{12}$ bacteria per gram of feces, bacteria can engage in competition over resources and colonization environments using an array of antagonistic mechanisms[2,3]. One prominent mechanism encoded by diverse Gram-negative bacteria including many highly abundant members of the gut microbiota, is the type VI secretion system (T6SS), which delivers effector proteins to recipient cells in a manner

dependent on prolonged cell contact[4]. The activity, regulation, structure, and ecological significance of the T6SS has been well-investigated through the work of many researchers primarily in representative species deriving from the phylum Pseudomonadota[5–7]. The minimum essential components of the Proteobacterial T6SS are 13 proteins, which make up three subcomplexes: the Membrane Complex (MC), Baseplate (BP), and Tail-Tube Complex (TTC)[8,9]. The 1.7 megadalton MC in Proteobacteria is formed by TssJ, TssL, and TssM, which create a five-fold symmetric transenvelope channel[10,11]. Anchored to the

[1]Laboratoire d'Ingénierie des Systèmes Macromoléculaires (LISM), Institut de Microbiologie, Bioénergies et Biotechnologie (IM2B), Aix-Marseille Université - Centre National de la Recherche Scientifique (CNRS), Unité Mixte de Recherche (UMR) 7255, Institut national de la santé et de la recherche médicale (INSERM), Marseille, France. [2]Department of Microbiology and Immunology, Geisel School of Medicine at Dartmouth College, Hanover, NH 03755, USA. [3]Institut Pasteur, Université Paris Cité, CNRS UMR3525, Microbial Evolutionary Genomics, Paris, France. [4]Department of Microbiology, University of Washington, Seattle, WA 98109, USA. [5]Microscopy Core Facility, Institut de Microbiologie de la Méditerranée (IMM), FR3479, CNRS, Aix-Marseille University, Marseille, France. [6]Department of Biology, University of Konstanz, Universitätsstraße 10, 78457 Konstanz, Germany. [7]Konstanz Research School Chemical Biology, University of Konstanz, Universitätsstraße 10, 78457 Konstanz, Germany. [8]Molecular Microbiology and Structural Biochemistry (MMSB, UMR 5086), CNRS & University of Lyon, 7 Passage du Vercors, 69007 Lyon, France. [9]Present address: Laboratoire de Chimie Bactérienne (LCB), Institut de Microbiologie, Bioénergies et Biotechnologie (IM2B), Aix-Marseille Université - Centre National de la Recherche Scientifique (CNRS), Unité Mixte de Recherche (UMR) 7255, Institut national de la santé et de la recherche médicale (INSERM), Marseille, France. ✉e-mail: benjamin.d.ross@dartmouth.edu; eric.durand@inserm.fr

cytoplasmically-protruding portion of the MC is the 1.15 megadalton BP, composed of TssE, TssF, TssG, and TssK, which form an outer layer around an inner hub of trimeric VgrG and monomeric PAAR[12]. Hcp subunits polymerize from the baseplate in hexameric rings, forming the tail of the TTC, which is encased in a sheath of repeating TssB and TssC subunits[13]. When activated, the sheath rapidly contracts, propelling the tail topped with VgrG/PAAR through the open pore of the MC, allowing for translocation of effector proteins[14].

From an evolutionary point of view, many components of the T6SS are sequence or structural homologs from bacteriophage (T4, Mu and Siphophage) components[15]. This is notably the case of the tail-tube complex, including the sheath protein sequences. Using the latter as a phylogenetic marker, T6SSs have been classified into four types, T6SS[i] through T6SS[iv], with further subclass divisions in T6SS[i][16]. According to these previous works, the T6SS[iv] has an independent origin and, among the others, the T6SS[iii] is the basal system in this group, *i.e.* the one emerging first from the last common ancestor of the T6SS[i-iii]. It is only found in the phylum Bacteroidota[16]. Mammalian gut symbionts of the order Bacteroidales possess three distinct T6SS genetic architectures (GAs), the best studied of which is GA3, encoded solely by *Bacteroides fragilis*[17]. Many but not all *B. fragilis* strains carry an intact GA3 locus[18–21]. Surprisingly, all three Bacteroidales GAs lack orthologs for the MC genes *tssJ, tssL*, and *tssM*, and have several uncharacterized genes, including five designated *tssN, tssQ, tssO, tssP*, and *tssR* in GA3[16,17]. Since it is unlikely that the T6SS could function without a MC, we predicted that the T6SS[iii] was likely to utilize a unique MC. Another possibility is that an alternative complex allowing passage of T6SS needle should exist in T6SS[iii].

The lack of a known MC for the T6SS[iii] led us to raise three questions: 1) Are the unknown Bacteroidota proteins TssNQOPR essential for T6SS[iii] function? 2) Do these proteins have the features expected for components of a MC? 3) What are the genomic patterns for the genes encoding TssNQOPR in relation to the other components of the T6SS[iii]? To answer these questions, we combined genetics, biochemistry, protein-protein interaction, structural modeling, fluorescence microscopy, and evolutionary genomics. These studies lead us to propose that the Bacteroidota T6SS[iii] has a unique architecture dictated by a MC composed of TssNQOPR.

## Results

### Identification of five genes encoding essential membrane-associated proteins of the *B. fragilis* T6SS[iii]

The *B. fragilis* T6SS[iii] operon has been identified and annotated previously[16,17,19–21]. Genes encoding the BP and TTC protein homologs have clearly recognizable homologs in the T6SS[iii] (Fig. 1a–b). However, all the components of the T6SS[i] MC lack homologs in the *B. fragilis* T6SS[iii] clusters (Fig. 1a–b) as well as the greater phylum[16]. A number of genes with unknown annotation and function are part of this putative transcriptional unit. Specifically, the *tssN, -Q, -O, -P* and *-R* genes (BF9343_1925, _1922, _1921, _1920 and _1919, respectively) are conserved in Bacteroides T6SS[iii] clusters.

Using bioinformatic analysis and structural predictions, we revealed that these five genes encode four transmembrane proteins and one lipoprotein (Fig. 1c, Supplementary Fig. 1). TssN presents five transmembrane helices and a globular cytoplasmic domain. TssQ, -O and -P each have a single N-terminal transmembrane helix followed by a periplasmic domain. TssR harbors an N-terminal SPII signal sequence specific to lipoproteins. We calculated the predicted structures of the TssNQOPR proteins using AlphaFold2 (see Methods)[22]. TssN harbors a five transmembrane helix (TMH) bundle connected to a globular cytoplasmic domain by a short linker (Fig. 1c). We searched for structural homologs using the DALI server and found that TssN held distant structural similarity to the T7SS protein EccD (PDBid 7B9F) from *Mycobacterium tuberculosis* (Supplementary Fig. 1c, d)[23]. TssN and EccD present the same topology, although EccD has ten TMHs. The overall fold of EccD's cytoplasmic domain superimposes well with the

predicted structure of TssN (residues 184-287, RMSD 2.68Å). TssQ and TssO are predicted to form elongated helical structures with little homology to known proteins, with the exceptions of a distant structural similarity for TssQ to the TIP20 protein, a central subunit of the SNARE complex that mediates membrane fusion in eukaryotes (PDB 6wc3, RMSD 2.5Å), and TssO harboring helical features similar to a DNA Damage-binding protein (PDB 7zn7, RMSD 2.8Å). TssP has a predicted four-domain architecture, including one TMH (1-32) and three contiguous periplasmic domains (D2-D4). Two of these latter domains (D2, 33-109 and D3, 110-192) are structurally similar to Polycystic Kidney Disease (PKD) domain of the collagenase from *Clostridium histolyticum* (PDB 2y72, RMSD 1.4Å). The predicted structure of TssR presents a compact and globular architecture consisting of three domains (D1, 1-231; D2, 277-558 and D3, 232-276 & 559-790). D1 and D3 have a broad interaction interface, whereas D2 is independent. TssR_D2 shows structural homology with the Von Willebrand Factor (vWA) domain (PDB 5BV8, RSMD 3.7Å). In conclusion, our AlphaFold2 modeling confirms the transmembrane helix topology of the five proteins determined above. This information allows us to build a membrane-associated model of the T6SS[iii] MC components, wherein TssN contributes a cytoplasmic hub while the four other proteins (TssQOPR) assemble a periplasmic architecture (Fig. 1c).

To test the necessity of these *tssNQOPR* genes for T6SS function, we constructed in-frame chromosomal deletion mutants in the *B. fragilis* NCTC 9343 strain (see Methods). We then performed bacterial competition assays against the susceptible target strain *B. thetaiotaomicron*. The co-culture assays revealed that the *tssNQOPR* genes are individually required for T6SS-dependent competition to the same extent as the gene encoding the TssC TTC subunit (Fig. 1d). Ectopic chromosomal complementation under constitutive promoters expressing each of the deleted genes individually restored the wild-type level of competitive index for each mutant, suggesting no polar effect of the gene deletion. To confirm the involvement of these genes in T6SS function, we monitored the release of the hallmark Hcp (TssD) protein in bacterial broth supernatants. Each of the five TssNQOPR proteins were separately found to be essential for Hcp secretion, similar to the TssC subunit (Fig. 1e). In conclusion, we have identified five putative membrane associated proteins, conserved in Bacteroidales, that are essential to the function of the *B. fragilis* T6SS[iii].

### TssNQOPR assemble a megadalton membrane complex in *B. fragilis*

We next sought to assess if TssNQOPR assemble a complex in the *B. fragilis* cell envelope. The *tssN* and *tssO* genes were tagged at their native loci in the *B. fragilis* NCTC 9343 genome to express endogenous levels of TssN[STREP] and [HIS]TssO fusion proteins, while retaining T6SS functionality (Supplementary Fig. 1b). Total membranes were isolated from cell pellets and solubilized using detergents. We performed a two-step affinity chromatography in order to recover only complexes and not unbound proteins. Mass spectrometry analysis of the purified sample identified the two baits, TssN[STREP] and [HIS]TssO, as well as TssQ, TssP and TssR as co-eluting binding partners, suggesting the formation of a five-protein complex (Fig. 2a). The Western blot confirmed the presence of the two baits TssN[STREP] and [HIS]TssO in the elution fraction, proving their reciprocal interaction (Fig. 2b). Interestingly, the estimated relative stoichiometry based on the emPAI calculated from the mass spectrometry data (Fig. 2c) revealed that TssN and TssQ are the most abundant proteins (n=5), followed by TssP (n=4), then TssR (n=2) and the less abundant TssO (n=1).

To confirm that the TssNQOPR proteins can form a complex independently of any other *B. fragilis* proteins, we constructed a plasmid-based system to overproduce the five proteins in *E. coli* BL21(DE3) (Supplementary Fig. 2). Constructs were designed to add STREP, FLAG, 8HIS, HA and VSVG tags at the carboxy (C) terminus of TssN, amino (N) terminus of TssQ, N terminus of TssO, N terminus of

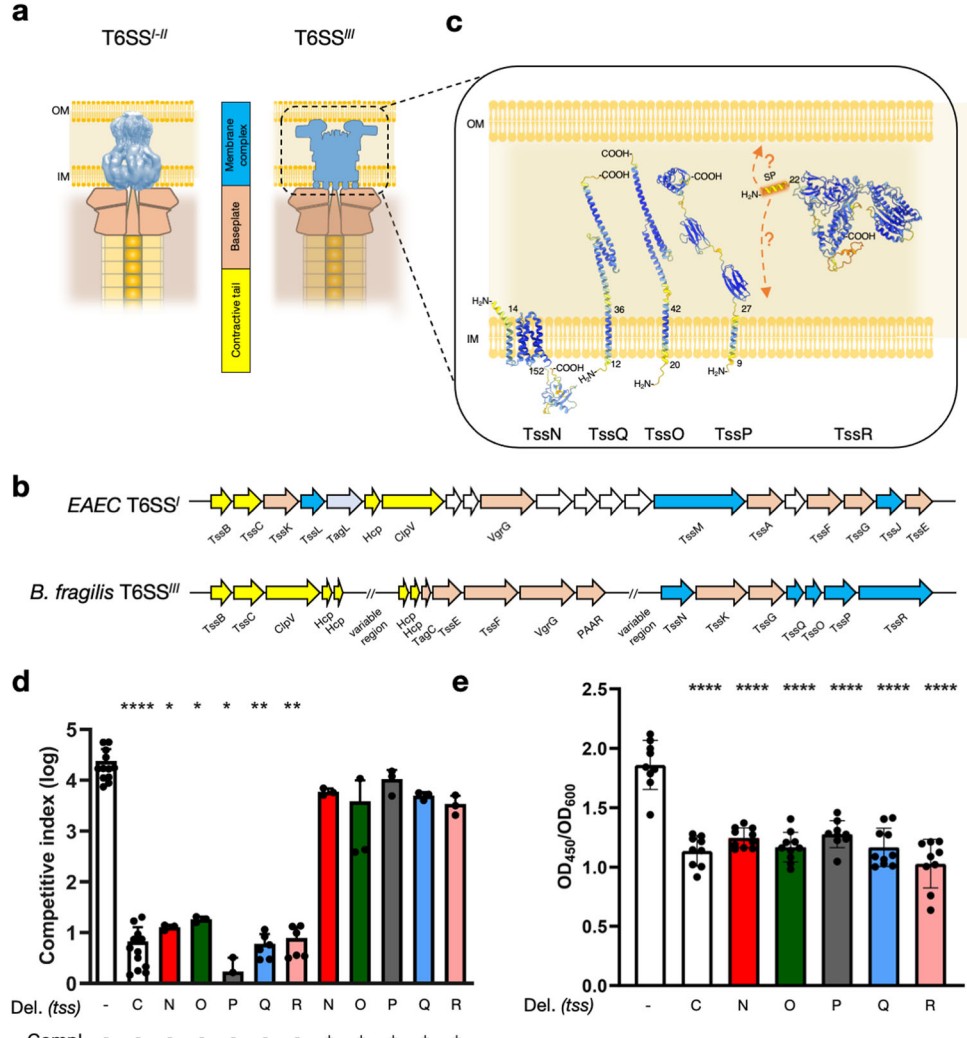

**Fig. 1 | The five genes *tssNQOPR* are essential for T6SS function in *B. fragilis*.**
**a** Schematic representation showing the difference between canonical T6SS and the *Bacteroides fragilis* membrane complex. **b** T6SS locus comparison between enteroaggregative *E. coli* and *B. fragilis* colored with the same color code used in panel (**a**). **c** AlphaFold2 models of the TssNQOPR proteins showing their topology relative to the membranes. **d** 16-hr anaerobic co-cultures of *B. fragilis* and *B. thetaiotaomicron* to measure T6SS-dependent competition. Competitive index calculated as the ratio of (donor *B. fragilis*/recipient *B. thetaiotaomicron*)$_{final}$ / (donor *B. fragilis*/recipient *B. thetaiotaomicron*)$_{initial}$ CFUs. In-frame chromosomal deletions of *tssN, tssO, tssP, tssQ*, and *tssR* respectively resulted in ablation of competitive advantage. Advantage was restored with complementation of chromosomal single

copy insertions of *tssN, tssO, tssP, tssQ*, and *tssR* respectively. * indicates *P* values = 0.037, ** = 0.004, **** ≤ 0.0001, two-sided unpaired t tests. Mean ± s.d. are shown; *n* = 12 for wildtype and Δ*tssC*, *n* = 6 for Δ*tssQ* and Δ*tssR*, and *n* = 3 for other strains, independent biological replicates that are each the mean of 3 technical replicates. **e** Anti-Hcp ELISA performed on the supernatants of corresponding *B. fragilis* strains to quantify levels of Hcp secretion. As in **a**, deletion strains reduced Hcp secretion levels to baseline. Results are normalized to cell density (OD$_{600}$). **** indicate *P* values ≤ 0.0001, one-way ANOVA. Mean ± s.d. are shown; *n* = 9 for wildtype, Δ*tssC*, Δ*tssR*, and Δ*tssP*, *n* = 10 for Δ*tssO*, Δ*tssN*, and Δ*tssQ*; independent biological replicates each with 3 technical replicates. Source data are provided as a Source Data file.

TssP, and C terminus of TssR, respectively. Total membranes were isolated and solubilized using detergent. One-step affinity chromatography followed by gel filtration using mild crosslinking (see Methods) resulted in the purification of a complex containing TssN, TssQ, TssO, TssP, and TssR (Fig. 2d). On the contrary, when the STREP tag on TssN was replaced by a HA tag, none of these five proteins were retained on the affinity chromatography column, proving the specificity of the isolated complex. Purified complexes were visualized by negative-stain electron microscopy (EM) (Fig. 2e). The TssNQOPR membrane complex assembles large and isolated particles about 230-254Å wide and 315-360Å long (Supplementary Fig. 5). In conclusion, the five TssNQOPR proteins encoded by genes within the *B. fragilis* T6SS$^{iii}$ genetic cluster interact to form a large membrane-embedded complex.

To gather more information on the structure of the assembly components, we identified structurally proximal amino-acids using

cross-linking mass spectrometry (XL-MS). The mildly stabilized TssNQOPR complex was purified and further linked with a NHS-ester reactive chemical crosslinker (DSSH12/D12). Two samples were analyzed, one (annotated: strep-Grafix) was mildly stabilized and objected to gel filtration previous to crosslinking, and another less processed sample (annotated: strep-only) that is expected to contain more non-specific binders and small subcomplexes. The purified TssNQOPR complex was subjected to cross-linking mass spectrometry analysis (XL-MS) and the resulting intra-protein cross-links were mapped on AlphaFold2 structures, for structural validation (Fig. 1, Supplementary Table 1, Supplementary Fig. 3, Source Data File 4). We obtained 64 high-confidence unique crosslinks, of which only 12% are violated, for which the distance of the two corresponding residues exceeds the maximal length of the crosslinker, mainly in TssQ (6 violations over a total of 18 crosslinks). The majority of cross-links are in agreement with

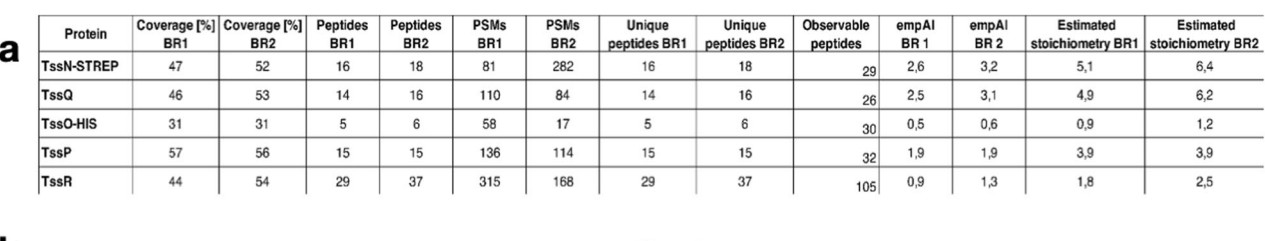

a

| Protein | Coverage [%] BR1 | Coverage [%] BR2 | Peptides BR1 | Peptides BR2 | PSMs BR1 | PSMs BR2 | Unique peptides BR1 | Unique peptides BR2 | Observable peptides | empAI BR 1 | empAI BR 2 | Estimated stoichiometry BR1 | Estimated stoichiometry BR2 |
|---|---|---|---|---|---|---|---|---|---|---|---|---|---|
| TssN-STREP | 47 | 52 | 16 | 18 | 81 | 282 | 16 | 18 | 29 | 2,6 | 3,2 | 5,1 | 6,4 |
| TssQ | 46 | 53 | 14 | 16 | 110 | 84 | 14 | 16 | 26 | 2,5 | 3,1 | 4,9 | 6,2 |
| TssO-HIS | 31 | 31 | 5 | 6 | 58 | 17 | 5 | 6 | 30 | 0,5 | 0,6 | 0,9 | 1,2 |
| TssP | 57 | 56 | 15 | 15 | 136 | 114 | 15 | 15 | 32 | 1,9 | 1,9 | 3,9 | 3,9 |
| TssR | 44 | 54 | 29 | 37 | 315 | 168 | 29 | 37 | 105 | 0,9 | 1,3 | 1,8 | 2,5 |

**Fig. 2 | *B. fragilis* assembles a TssNQOPR complex. a** Identification of the 5 proteins using mass spectrometry after a pulldown using TssN-STREP bait, two biological replicates (BR) were performed, and exponentially modified protein abundance index (emPAI) is calculated. **b** Western blot α-STREP and α-HIS showing the purification of $^{STREP}$TssN and $^{H}$TssO after the pulldown using the STREP tag of TssN as a bait. **c** Estimated stoichiometry based on the emPAI calculated from the mass spectrometry analysis. Data are presented as mean values +/- SD. **d** Immunoblotting showing the copurification of the TssNQOPR complex using TssN-STREP as a bait followed by SEC chromatography. Load (L) and Elution (E) were loaded on a 12.5%-acrylamide SDS PAGE, and immunodetected with respective anti-tag antibody. The position of the proteins is indicated on the right, molecular weight markers (in kilodaltons) are indicated on the left. The negative control shows no copurification. **e** Negative staining micrographs of the TssNQOPR complex purified from *E. coli* overproduction. For western blots and micrographs, representative images from triplicate experiments are presented.

the predicted structures. Furthermore, we observe that 16 satisfied crosslinks map on amino acids that are at least 20 residues apart in the sequence, thereby validating the underlying tertiary structure. The violated cross-links in TssQ might arise from inter-molecular proximity occurring in the complex within two copies of the same proteins.

**An intricate protein-protein interaction network assembles the TssNQOPR complex**
To understand the mode of assembly of the TssNQOPR complex, we sought to decipher its protein-protein interaction (PPI) network. We thus systematically performed pairwise copurifications assays from heterologous expression in *E. coli*. For this purpose, additional plasmids were constructed (see Methods) to be able to monitor all combinations of binary interaction. Pairs of protein candidates were then produced in *E. coli* from the chosen combination of plasmids, extracted either from membranes with detergent or directly from the soluble compartments, and affinity purified. These experiments were performed in replicate trials with reproducible results. Following this approach, we demonstrated that the bait TssN (TssN$^{S}$) is able to pull-down TssQ, TssO and TssR, but not TssP (Fig. 3 and Supplementary Fig. 4a). On the contrary, TssQ, TssO and TssR are not purified from the STREP-affinity column in the absence of TssN (Supplementary Fig. 4b),

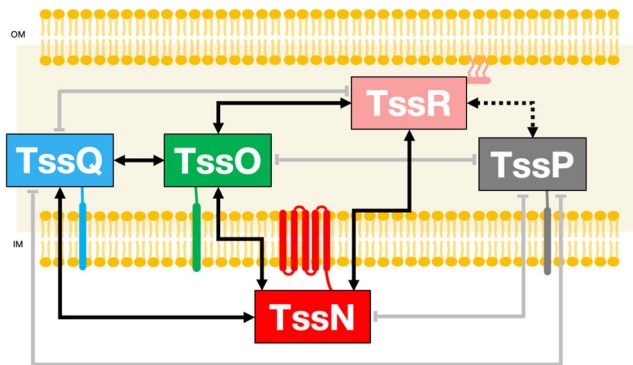

**Fig. 3 | Interaction network in the complex.** Schematic representation of the interactions between all of the TssNQOPR proteins using copurification. The full arrows represent copurification observed in full-length proteins, the dashed arrow represents copurification of the periplasmic truncations of the proteins. A grey arrow with a flat end indicates that no co-purification was observed.

proving the specificity of the interactions. The bait TssQ (TssQ$^{GST}$) is not able to pull-down TssP and TssR (Supplementary Fig. 4c). The periplasmic domain of TssR (TssRp$^S$) is able to pull-down the periplasmic domain of TssP (TssPp) (Supplementary Fig. 4d). On the contrary, TssPp is not purified from the STREP-affinity column in the absence of TssRp (Supplementary Fig. 4e), proving the specificity of the TssRp-Pp interaction. The bait TssO (TssO$^H$) is able to pull-down TssQ and TssR but not TssP (Supplementary Fig. 4f), whereas TssQ and TssR do not bind to the HIS-affinity column alone (Supplementary Fig. 4g), proving the specificity of the TssO-R interaction. In summary, the assembly of the *B. fragilis* TssNQOPR membrane complex involves an intricate PPI network wherein TssN, TssO and TssR interact with each other, TssQ interacts with TssN and TssO, and TssP only interacts with TssR (Fig. 3).

### The TssNQOPR complex is connected to the baseplate
TssN appears to play a central role in the PPI network to facilitate assembly of the TssNQOPR complex. We wondered if TssN could play a "hub" function in the membrane complex. For this reason, we investigated the oligomeric state of TssN. Purified TssN in detergent micelles forms large homo-multimers as observed by DLS (Fig. 4a, b) and EM-NS observation (Fig. 4c). To test the importance of TssN TMH in multimer assembly, we cloned and expressed the isolated cytoplasmic domain (TssN$_{Cyto}$). Production assays in *E. coli* show that TssN full length is capable of forming SDS-resistant oligomers, but not TssN$_{Cyto}$ (Fig. 4d), demonstrating that TssN helices are involved in homoligomerization.

To further demonstrate that the TssNQOPR complex is the T6SS$^{iii}$ MC, we investigated its connection with the conserved BP complex. Through its cytoplasmic domain, TssN possesses the only available interface to interact with the BP. TssK is the central component of the BP and in the T6SS$^i$ it interacts with the canonical TssJLM MC[24,25]. We thus performed co-production of TssN$^S$ and *B. fragilis* TssK (BF9343_1924) in *E. coli* and tested their interaction by co-purification (Fig. 4e). Although TssK alone is not eluted from the STREP-affinity column, TssN$^S$ is able to co-purify TssK. In contrast, TssQ, which does not possess a significant cytoplasmic domain, does not co-purify with TssK. This further validated TssN as the TssNQOPR hub protein that interacts with the BP in vitro.

To confirm the "in vivo" relevance of this interaction and the key role of TssN$_{Cyto}$ in recruiting the BP, we overproduced TssN$_{Cyto}$ in *B. fragilis*. We cloned *B. fragilis* strains expressing ectopic chromosomal TssN$_{Cyto}$ in wildtype or *tssN* deletion backgrounds. To investigate the impact of TssN$_{Cyto}$ on MC stoichiometry, we employed two different promoters, a "low" constitutive promoter (BT1311), and a "high"

constitutive promoter (P1T$_{DP}$$^{A21}$), with previously validated expression levels[26]. These strains were assayed as before in competition with *B. thetaiotaomicron* (Fig. 4f). Expression of TssN$_{Cyto}$ in the *tssN* deletion background ablated *B. fragilis* competition, proving the cytoplasmic domain of TssN is not sufficient for T6SS-mediated competition. Low expression of TssN$_{Cyto}$ in a strain expressing full-length TssN displayed no competitive disadvantage, while high expression of TssN$_{Cyto}$ in the wildtype strain reduced competition, indicating that the cytoplasmic domain of TssN can inhibit T6SS function in a manner likely dependent on stoichiometry. In conclusion, we demonstrated that TssN acts as a hub in the *B. fragilis* T6SS, nucleating the assembly of the MC and recruiting the BP through interactions with its cytoplasmic domain.

### Proper *B. fragilis* sheath assembly requires TssNQOPR
We next sought to determine if TssNQOPR contributed to the assembly of other T6SS$^{iii}$ subcomplexes. One such subcomplex is the sheath, which has been monitored as a readout of comprehensive T6SS assembly since proper sheath polymerization requires both membrane complex and baseplate formation[25]. We constructed superfolder-GFP (sfGFP) fused TssB overexpression strains in a *B. fragilis tssB* deletion background[27]. As a negative control, we also included a *tssK* deletion strain. "Wildtype" *ΔtssB* with TssB-sfGFP overexpression displayed either extended or contracted TssB-sfGFP sheaths in approximately 5% of cells, a frequency similar to that found for other species (Fig. 5a). As expected, the *tssK* deletion strain exhibited a significant inability to polymerize sheaths. Similarly, *ΔtssNQOPR* strains each displayed reductions in the quantity of cells with sheaths, though the *tssR* deletion was not statistically significant compared to wildtype (Fig. 5b, c). We next investigated the subcellular localization of sfGFP-TssB in WT and mutant strains. In the wildtype and *ΔtssN* strains, sfGFP-TssB foci were uniformly distributed across the cell periphery. In contrast, the *ΔtssQ, ΔtssO, ΔtssP*, and *ΔtssR* mutants exhibited a shift in localization of TssB-sfGFP to the poles of the cell (Fig. 5d, e), a pattern which has been reported for general protein aggregates in other species of bacteria[28]. We also quantified the length of sfGFP-TssB foci in all strains. This analysis revealed that the distribution of sheath lengths also skewed significantly longer in MC mutants compared to the wildtype strain, possibly indicating that the few sheaths that can form in the absence of a functional MC may be detached from the baseplate, as observed in other contexts (Fig. 5f)[29]. In summary, we find that, in addition to a requirement of TssNQOPR for T6SS function, the MC proteins are required for the assembly and function of other T6SS subcomplexes.

### Evolution and diversity of the T6SS$^{iii}$
We investigated the presence of the MC components TssNQOPR in other T6SS$^{iii}$. We searched for the T6SS$^{iii}$ using improved TXSScan models and protein profiles in 1253 complete genomes of Bacteroidota[30]. These models define the components of the T6SS, their quorum, and their genetic organization (see Methods). We identified 443 T6SS$^{iii}$ in the Bacteroidota and none in more than 20,000 genomes of other Bacteria. We then searched for loci encoding the MC components in the absence of T6SS to identify the putative ancestral system that was co-opted to produce the MC. We found no genome with a complete MC and lacking all other genes of the T6SS; in all cases the number of genes for the MC was lower than that of the other 9 core components of the T6SS$^{iii}$. Of note, no genome in Bacteroidota had a T6SS$^i$-like MC. Hence, the T6SS$^{iii}$ is present only in Bacteroidota and is the only type found in the phylum. The MC (or part of it) is found in all these systems and is specifically associated with the presence of the T6SS$^{iii}$.

To assess the taxonomic distribution of the T6SS$^{iii}$, we built a rooted phylogeny of the Bacteroidota using a set of nearly ubiquitous genes in the phylum (Supplementary Fig. 6). We also built a smaller tree with just one genome per species for visualization purposes

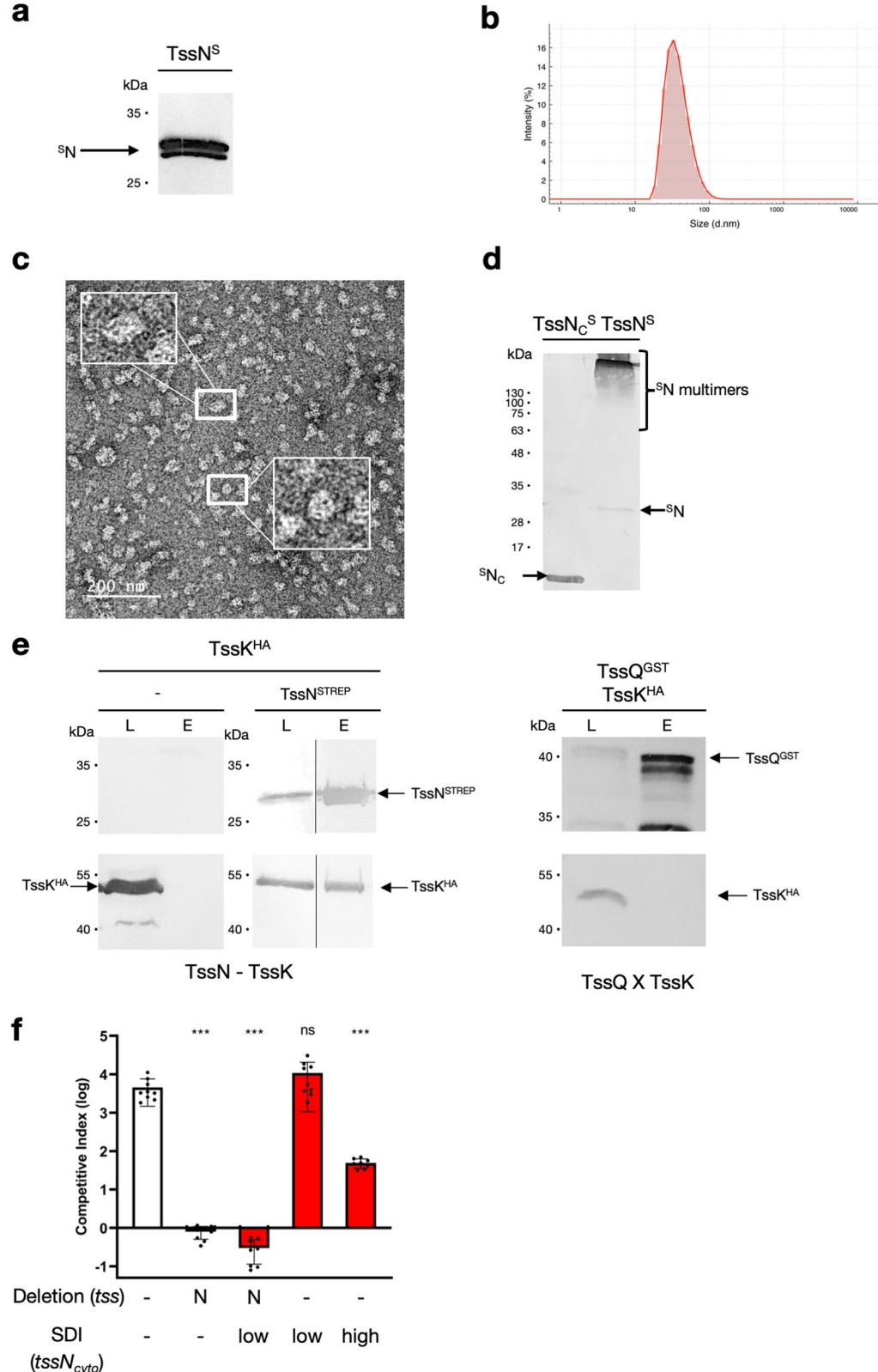

(Fig. 6, see Methods). These analyzes revealed that 149 species have a T6SS[iii], which are present in 44 genera (out of 174). These values should be regarded as an under-estimate since some genera are very poorly sampled (and many others are probably not yet known). Interestingly, T6SS[iii] are unevenly scattered across the phylum, with some genera having many systems whereas others lack them altogether. Notably, the clades of obligate mutualistic endosymbionts with small genomes including *Sulcia*, *Karelsulcia*, and *Blattabacterium* lack T6SS[iii]. In contrast, the distribution of T6SS[iii] is quite ubiquitous in some genus, e.g. *Chryseobacterium* and *Elizabethkingia*, but most often only a fraction of the species encodes T6SS[iii], like in *Flavobacterium*, *Bacteroides*, *Prevotella*, *Mucilaginibacter* or *Spirosoma*. The systems lacking some components of the MC are concentrated in *Spirosoma*, *Hymenobacter* (which lacks any complete

**Fig. 4 | Interaction with the baseplate. a** Immunoblotting showing the purification of the TssN-S homomultimer. **b** Dynamic Light Scattering graph showing the average size of the TssN-S homomultimers. **c** Electron micrograph showing the multimers formed by purified TssN-S. Insets show zoom-in of TssN multimer particles. **d** Comparison of extracts of *E. coli* strains producing TssN-S and TssNc-S loaded on a 12,5% acrylamide SDS PAGE and immunodetected by anti-strep antibody. **e** Immunoblotting showing the copurification of the TssK$^{HA}$ with TssN$^S$, purified with a STREP-trap affinity column. Load (L) and Elution were loaded on a 12.5%-acrylamide SDS PAGE, and immunodetected with respective anti-tag antibody. The position of the proteins is indicated on the right, molecular weight markers (in kilodaltons) are indicated on the left. The TssQ$^{GST}$-TssK$^{HA}$ copurification is used as a negative control and shows no copurification. **f** Anaerobic co-cultures of

donor *B. fragilis* and recipient *B. thetaiotaomicron* to measure T6SS-dependent competition. Competitive index calculated as the ratio of donor/recipient$_{final}$ / donor/recipient$_{initial}$ CFUs. Strains are combinations of wildtype *tssN*, in-frame chromosomal deletions of *tssN*, and chromosomal single copy insertions of *tssN$_{cyto}$* under constitutive highly expressing promoter (P1T$_{DP}$$^{A2l}$) or constitutive moderately expressing promoter (BT1311). *** indicate *P* values < 0.0001, ns not significant, two-sided unpaired t tests. Mean ± s.d. are shown; *n* = 8 for *ΔtssN* TssNc high, *n* = 9 for others; independent biological replicates that are each the mean of 3 technical replicates. For western blots and micrographs, representative images from triplicate experiments are presented. Source data are provided as a Source Data file.

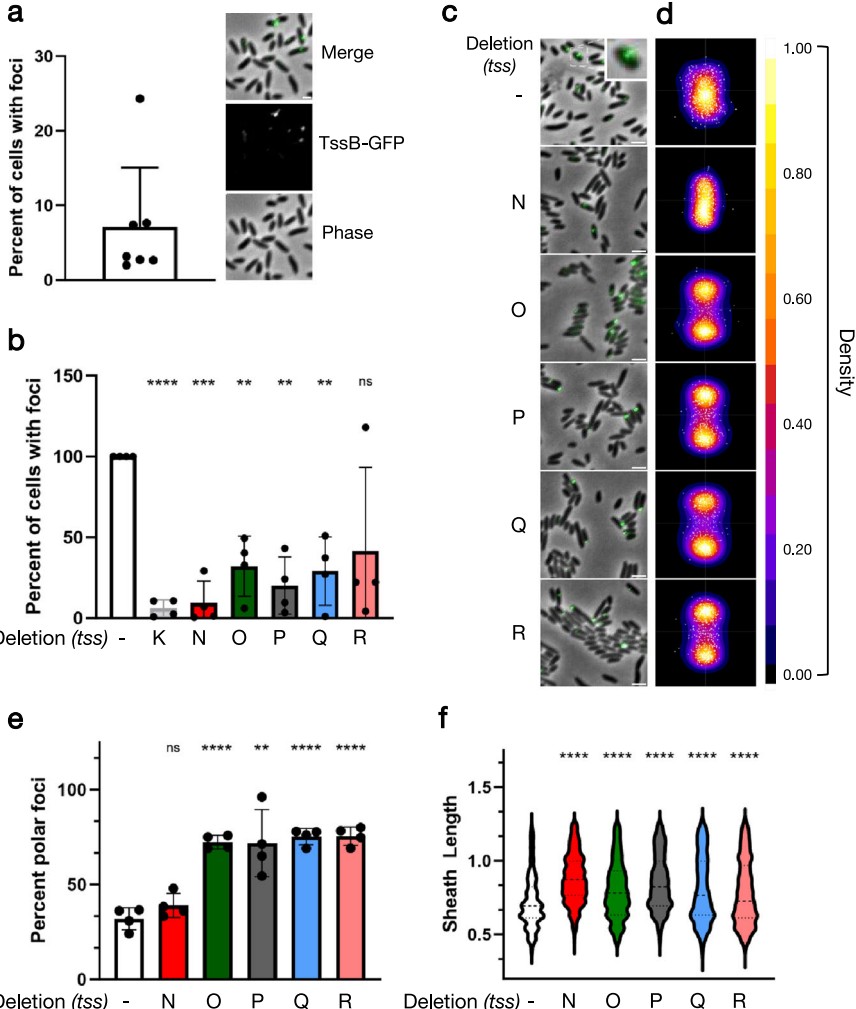

**Fig. 5 | TssNQOPR are required for proper T6SS$^{iii}$ sheath assembly in *B. fragilis*. a** Percent of wildtype *B. fragilis* cells expressing TssB-GFP sheaths. Mean + s.d. is shown; each dot represents the number of foci / number of cells with TssB-GFP foci in a biological replicate. Seven biological replicates with four fields (technical replicates) per biological replicate were measured, totaling 93887 cells and 5293 foci. Representative field of cells included as an inset, with phase contrast, TssB-GFP fluorescence, and merged composite micrographs. **b** Percent of *B. fragilis* cells with TssB-GFP foci, normalized to the corresponding wildtype strain per biological replicate. *n* = 93887 wildtype, 46235 *ΔtssK*, 27668 *ΔtssN*, 54976 *ΔtssO*, 28979 *ΔtssP*, 39085 *ΔtssQ*, 46603 *ΔtssR*, cells analyzed; *n* = 5293 wildtype, 120 *ΔtssK*, 484 *ΔtssN*, 1233 *ΔtssO*, 660 *ΔtssP*, 802 *ΔtssQ*, 1009 *ΔtssR* foci analyzed. ns indicates *P* values = 0.109, ** indicates *P* value = 0.007 for *ΔtssO*, 0.003 for *ΔtssP*, and 0.005 for *ΔtssO*, *** indicates *P* value = 0.0009, **** indicates *P* value < 0.0001. **c** Representative

composite micrographs of strains quantified in (**b**), merge of phase contrast and TssB-GFP fluorescence. Additional inset of an individual cell elaborating an extended TssB-GFP sheath shown for wildtype in white box. **d** Subcellular localization heatmaps generated from ~500 random TssB-GFP foci per strain. Density of TssB-GFP foci plotted with heatmap LUT and as individual white foci. Strains correspond to labels in (**c**). **e** Percent of "polar" localized TssB-GFP foci relative to total foci per strain. ns indicates *P* values = 0.155, ** indicates *P* value = 0.005, **** indicates *P* value < 0.0001. **f** Distribution of TssB-GFP foci lengths across *B. fragilis* strains. **** indicates *P* value < 0.0001. Two-sided one sample t-test for (**b**) two-sided unpaired t-tests for **e** and **f**. Means ± s.d. are shown; *n* = 4 independent biological replicates that are each the mean of 4 technical replicates. Scale bars, 1 μm in (**a**), 2 μm in (**c**). Source data are provided as a Source Data file.

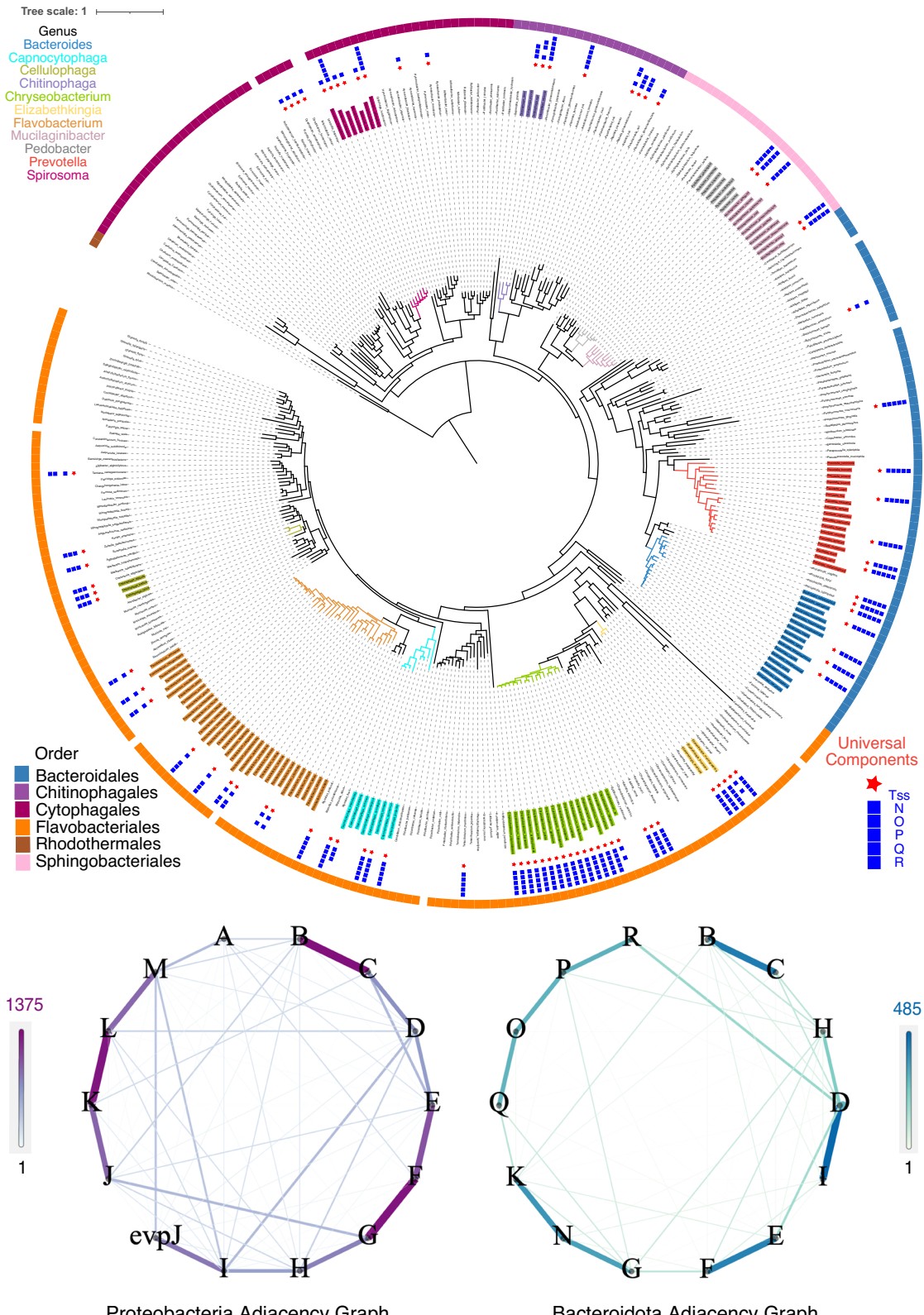

**Fig. 6 | Distribution and organization of T6SS^iii loci across Bacteroidota.** The tree of the phylum with one genome per species. The outer layers indicate the presence or absence of the T6SS (universal markers in just one group) and each of the MC components (see inset legend). The outer ribbon indicates the Order. The most represented genera are colored according to the colors indicated in the inset legend. Graphs depict the genetic organization of the T6SS of Proteobacteria (left) and Bacteroidota (right).

T6SS[iii]), and *Flavobacterium*. Two components – *tssN* and *tssP* – were nearly ubiquitous, whereas *tssO, tssQ* and *tssR* were sometimes absent or unidentifiable (resp. 27%, 23%, and 31%, Supplementary Fig. 7a). The biological significance of the absence of these components in poorly sampled species is hard to assess, since genomes may encode defective protein secretion systems[18,31]. Yet, we had many genomes of Flavobacteriales and we still often failed to identify *tssO, tssQ*, and *tssR* in the many T6SS of the clade. To check if we might be missing these components because of high sequence divergence, we took advantage of the fact that they tend to be encoded next to *tssP* (see below), which is nearly ubiquitous and conserved in sequence. Hence, we picked 10 genes around *tssP* when *tssOQ* were missing, folded these proteins using ESMfold, and searched for structural homologs of *tssO* and *tssQ* using foldseek (RMSD<3, see Methods)[32]. We could identify a large number (50) of additional putative TssOQ components in this way (mostly in *Flavobacterium*). While these results should be interpreted with care, as we are searching for homology among pairs of predicted small protein structures, many of the missing TssOQ could be false negatives caused by high sequence divergence.

It was previously shown that T6SS[iii] loci of Bacteroidales can be found in three different genetic architectures[17]. Our analysis of the genetic organization of the Bacteroidota T6SS[iii] loci revealed much more diverse genetic configurations at the level of the phylum. These precluded the definition of a few archetypes. Furthermore, the genes encoding the T6SS[iii] are often split in multiple loci scattered in the genome (Supplementary Fig. 7b). Notably, the systems of *Chryseobacterium* were split into up to four, and sometimes five, loci. To assess the variability of the genetic organization across the phylum, and identify groups of co-localized genes, we built a contiguity graph. Briefly, the different components of the T6SS[iii] were represented as nodes and the edges connect pairs of contiguous components in the genome (as long as they are at less than a maximal distance of 10 genes, see Methods). This revealed some groups of co-localized genes (Fig. 6). Interestingly, these groups of genes match key physical interactions between proteins in the T6SS. These include the pairs *tssDI* which together form the Hcp tube and its puncturing device, the pairs *tssBC* which together form the sheath, and *tssFE* which are both parts of the baseplate wedge. The MC components were usually separated in two parts. On one side, there is a group of four components *tssQOPR*, where *tssR* is often contiguous with *tssD* (itself often present in multiple copies). On the other side, we systematically found *tssN* between *tssK* and *tssG*. This parallels our experimental results above where TssN was shown to physically interact with TssK. To establish a comparison of these genetic organizations with those found in Proteobacteria, we built a similar graph for the T6SS[i]. It shows some common patterns, notably the pairs *tssBC* and *tssEF* are conserved. Proteobacteria also have a group of four contiguous components *tssjKLM* of which TssJLM are the Proteobacterial MC and TssK establishes the link with the phage-like complex[24]. Hence, whereas the MC connector to the BP is encoded in the neighborhood of the remaining MC in Proteobacteria, it is encoded next to the BP components in Bacteroidota. Overall, our results show that while the exact genetic organization of the T6SS is very variable across the phylum, there are conserved groups of genes and these encode proteins that interact physically within the system.

## Discussion
### Composition and topology of the components of the membrane complex
In this study, we have shown that the T6SS[iii] MC is composed of four inner-membrane proteins (TssNQOP) and one possible outer-membrane lipoprotein (TssR). In comparison, the T6SS[i] MC core is composed of two inner-membrane proteins (TssL and TssM) and one outer-membrane lipoprotein (TssJ)[10]. Such discrepancy in the number

of proteins that participate in the building of the MC could be explained by the different approaches that have been used to characterize the composition of these transenvelope complexes. In T6SS[i], a top-down approach based on sequence homology, structural predictions and sub-cellular localization has first defined TssJLM as the core of the megadalton complex[10]. Accessory components have been identified that help the building and positioning of the T6SS' MC in the bacterial cell envelope. In Enteroaggregative *E. coli* (EAEC) for instance, TagL comprises an N-terminal domain that mediates contact with TssM and a peptidoglycan-binding domain that binds the cell wall[33]. In addition, the EAEC MC has domesticated the protein MltE, a housekeeping transglycosylase that helps the crossing of the bacterial cell wall to accomplish the last steps of the MC assembly[34]. In our investigations of the T6SS[iii], we implemented a bottom-up approach that led to the identification of most of the components of the MC. Consequently, we propose that the 5-component T6SS[iii] MC could encompass both core as well as accessory factors. Another possibility is that our stringent biochemical pull-down assays from *B. fragilis* cells did not allow the isolation of loosely bound accessory components. Other complementary approaches will be needed in the future to identify any other protein partners. In particular, the mechanism by which the T6SS[iii] MC crosses the PG layer in the *B. fragilis* envelope is unknown.

Our genome analysis showed that the MC components are obligate features of the T6SS[III], confirmed that they are specific to Bacteroidota, and that this is the only type of MC found in the phylum. Hence, the MC seems to have arisen very early in the natural history of the phylum. Given the necessity of a MC for T6SS function, this process of co-option may have been the founding event in the evolutionary biogenesis of the T6SS[III]. The lack of any sequence homology between T6SS[i] and T6SS[III] MC suggests the two MC had independent evolutionary histories. The present data suggests three possible scenarios: an ancestral T6SS with a T6SS[i] MC that was replaced by the T6SS[iii] MC, an ancestral T6SS with a T6SS[iii] MC that was replaced by the T6SS[i] MC in Proteobacteria, or an ancestral T6SS that acquired a MC independently in the two phyla. In any case, these co-option events have shaped the MC composition and evolved convergently to produce the same complex function: to translocate effector proteins in a contact-dependent manner. Remarkably, the overall domain architecture and topology has been conserved: an outer-membrane lipoprotein that anchors the MC to the cell envelope, several periplasmic domains that assemble a 'cage' delineating the T6SS channel, several TM domains across the inner membrane, which then transmit the information to a cytoplasmic hub that docks the BP. One can envisage that these are the key fundamental aspects necessary for a functional T6SS membrane channel.

### Overall architecture of the T6SS membrane complex
The T6SS[i] MC core is 325Å in height and 209Å in diameter, dimensions that are compatible with the crossing of the bacterial envelope[10]. Interestingly, the T6SS[iii] MC observed on the electron micrographs appears to be at maximum 360Å in height and 250Å in width. Since the resolution of the two observations are different, we cannot strictly compare the sizes of these complexes but at least we can confidently assume that the T6SS[iii] MC possesses the dimensions necessary for its function as a T6SS channel. Future works will focus on the determination of the high-resolution structure of the T6SS[iii] MC to precisely measure its dimensions, the location of the various proteins and to delineate the channel that guides the Hcp tube/VgrG spike upon sheath contraction.

### Docking of the conserved T6SS baseplate on the Bacteroidota-specific membrane complex
The T6SS baseplate, composed of TssKFGE proteins, makes multiple interactions with the T6SS[i] MC[25]. Notably, TssK interacts with the TssM and TssL cytoplasmic domains. Surprisingly, the T6SS[iii] MC projects

only one domain in the cytoplasm (belonging to TssN). This raises the question of how the baseplate connects with the MC in Bacteroidota. In this study, we demonstrated that the baseplate protein TssK is recruited to the TssN cytoplasmic domain, making a unique and specific interface between the T6SS baseplate and the membrane complex. This key protein interaction is matched by the frequent co-localization of the two genes across Bacteroidota, producing a highly conserved group of *tssJ-K-L-M* genes in the T6SS[i] clusters (Fig. 6). The present identification of the synteny tssG-K-N-(QOPR) in T6SS[iii] clusters confirms the importance of the TssK-TssG as a determinant of the BP-MC[iii] connection in Bacteroidota. It remains to be determined how the TssK structure adapted to interact with TssN. It will then be interesting to compare this process with the one leading to the interactions of TssK with TssL and TssM in T6SS[i].

Although we demonstrated that TssK is recruited solely by TssN (Fig. 4), the five TssNQOPR proteins are all separately required for wildtype levels of sheath assembly (Fig. 5). This requirement could be explained by the intertwined protein-protein interaction network that we have identified and that connects the 4 other MC[iii] proteins to TssN (Fig. 3), suggesting a mutual stabilization as a prerequisite to assemble a functional baseplate. In *EAEC*, the MC protein TssM is required for the assembly of TssK-baseplate foci[25]. However, it is not known if the mere presence of TssM or its final assembly into the MC together with TssL and TssJ are the determinant for BP assembly. Our work here on the Bacteroidota T6SS[iii] suggests that the complete assembly of the MC is needed for functional BP docking.

## Predictive structural homology of the MC[iii] proteins

To learn more about the five membrane-associated proteins that form the Bacteroidota MC, we used AlphaFold2 to predict their 3D structures (Fig. 1 and Supplementary. Fig. 1). The structures were validated using XL-MS analysis of the assembly. The structures highlighted the structural homology of TssN with the protein EccD from Actinobacteria T7SSa. TssN and EccD share the same topology consisting of TM helices connected to a cytoplasmic domain[35]. This latter domain in EccD assembles a chamber made by a ubiquitin-like fold, also called the β-grasp fold. This fold is found in proteins having a strikingly diverse range of biochemical functions and notably is typical for scaffold proteins in macromolecular assemblies[36]. On the membrane side, EccD assembles as a secretion pore that is entirely hydrophobic and plugged with lipids[37]. It was proposed that insertion of substrates, which would need to expel the lipids inside the pore, could prime the EccD pore for transport by inducing conformational changes[35,38]. The EccD multimer facilitates extensive protein-protein interactions with the other core complex proteins. In a similar manner, we demonstrated that TssN assembles large homo-oligomers in the inner membrane (Fig. 4a–d), that it lies at the center of a complex PPI network (Fig. 3) and that it interacts with the TssK BP protein (Fig. 4e, f). These findings, combined with the structural homology with EccD, provide support for the hypothesis that TssN could represent the core of the T6SS[iii] MC. TssN would then be a central scaffold for building the MC, a possible route for substrate (Hcp/VgrG) passage, and a hub using its ubiquitin-like domain to dock the cytosolic BP (Fig. 7).

TssP is predicted to encompass two domains that are structurally similar to the PKD domain, whereas TssR_D2 domain shows structural homology with the Von Willebrand Factor (vWA) domain (Supplementary Fig. 1). Interestingly, these two types of domains are usually found in extracellularly exposed proteins such as collagenase (PKD) or extracellular matrix protein (vWA)[39,40]. One can speculate that massive conformational changes, like the ones anticipated during the tail-tube ejection, would trigger the opening of the MC leading to the temporary exposure of these domains at the exterior of the cell. Future work will try to elucidate the function of these domains in T6SS[iii] or elsewhere, which could be protein binding of the prey cell, adhesion to eukaryotic matrix or other possibilities.

We have identified unappreciated diversity in the genomic organization of T6SS[iii] loci. Interestingly, even though these loci were sometimes split, the patterns of contiguity between genes provide information on the interaction between proteins. It is well known that pairs of interacting proteins tend to form evolutionary stable genetic loci[41]. Here, we found that pairs of conserved gene neighborhoods match several of the key interactions in the T6SS[iii]. The reasons for this may be multiple: history (unlikely to be determinant, given the diversity of patterns for other pairs of genes), co-translation of gene transcripts may facilitate the assembly of the complex[42], and gene neighborhood could facilitate co-evolution of protein-protein interfaces due to genetic linkage that might otherwise be broken by homologous recombination.

We found the T6SS[iii] to be present in around a third of the genomes scattered across the tree of the Phylum Bacteroidota. The observed divergence in protein sequence and genetic organization of these systems suggests that they are very old. The sparse frequency of T6SS[iii] across most clades of the Bacteroidota suggests that they were frequently horizontally transferred and lost (or both). Horizontal transfer and evolutionary loss appear to be common among the T6SS[iii] of Bacteroidales[31]. Yet, the multi-loci organization of the genes encoding the system in some clades, notably in *Chryseobacterium*, suggests that in certain clades the T6SS cannot be easily transferred anymore. Since they tend to be ubiquitous in *Chryseobacterium* they are probably not lost either. This is reminiscent of studies on type IV filaments showing that systems that are rarely lost tend to be encoded in multiple loci, which then affects their ability to further horizontal gene transfer[43].

Our HMM-based genome scans failed to identify three MC components (*tssO*, *tssQ*, *tssR*) in a significant number of systems. Their absence seems to cluster in a few clades in the tree suggesting that their absence is not caused by recent pseudogenization events. Our preliminary comparisons of the predicted structures of the genes neighboring *tssP* led to the identification of putative homologs of TssO and TssQ, suggesting that in many cases these components may well be present in the system. Alternatively, it is possible that other genes were co-opted to the MC for the same function. Further work will be needed to understand how horizontal gene transfer and gene loss explain the distribution (and composition) of T6SS[iii] across the phylum. This may be key to unraveling the natural history of the MC, whose origin is difficult to identify at this stage, since no locus independent of the T6SS[iii] encodes this set of genes. The identification of the origin of the MC and of the co-option events leading to the emergence of the T6SS[iii] will therefore greatly benefit from the elucidation of its structure and its comparison with that of the T6SS[i].

## Methods

### Bacterial strains and growth conditions

The strains, plasmids, and oligonucleotides used in this study are listed in Supplementary Table 2. The *E. coli* K-12 DH5α strain was used for cloning procedures; the *E. coli* K-12 BL21(DE3) strain was used for protein expression and purification. Strains were routinely grown in lysogeny broth (LB) rich medium with shaking at 37 °C. Plasmids were maintained by the addition of streptomycin (100μgml−1), kanamycin (50μgml−1), ampicillin (100μgml−1) chloramphenicol (30μgml−1) or ampicillin (100μgml−1). Expression of genes from pCDF, pRSF, and pETDuet vectors was induced with 1mM of isopropyl-β-D-thio-galactopyrannoside (IPTG, Eurobio) for 16 hours at 16 °C. *B. fragilis* strains were grown on brain heart infusion (BHIS) agar plates or BHIS broth medium supplemented with 1 ugml-1 of vitamin K3 (Acros Organics/Fisher Scientific) and gentamicin (60 ugmL-1) under anaerobic conditions (10% $CO_2$, 10% $H_2$, 80% $N_2$) within a Whitley A55 anaerobic chamber (Don Whitley Scientific, Victoria Works, UK) at 37°C.

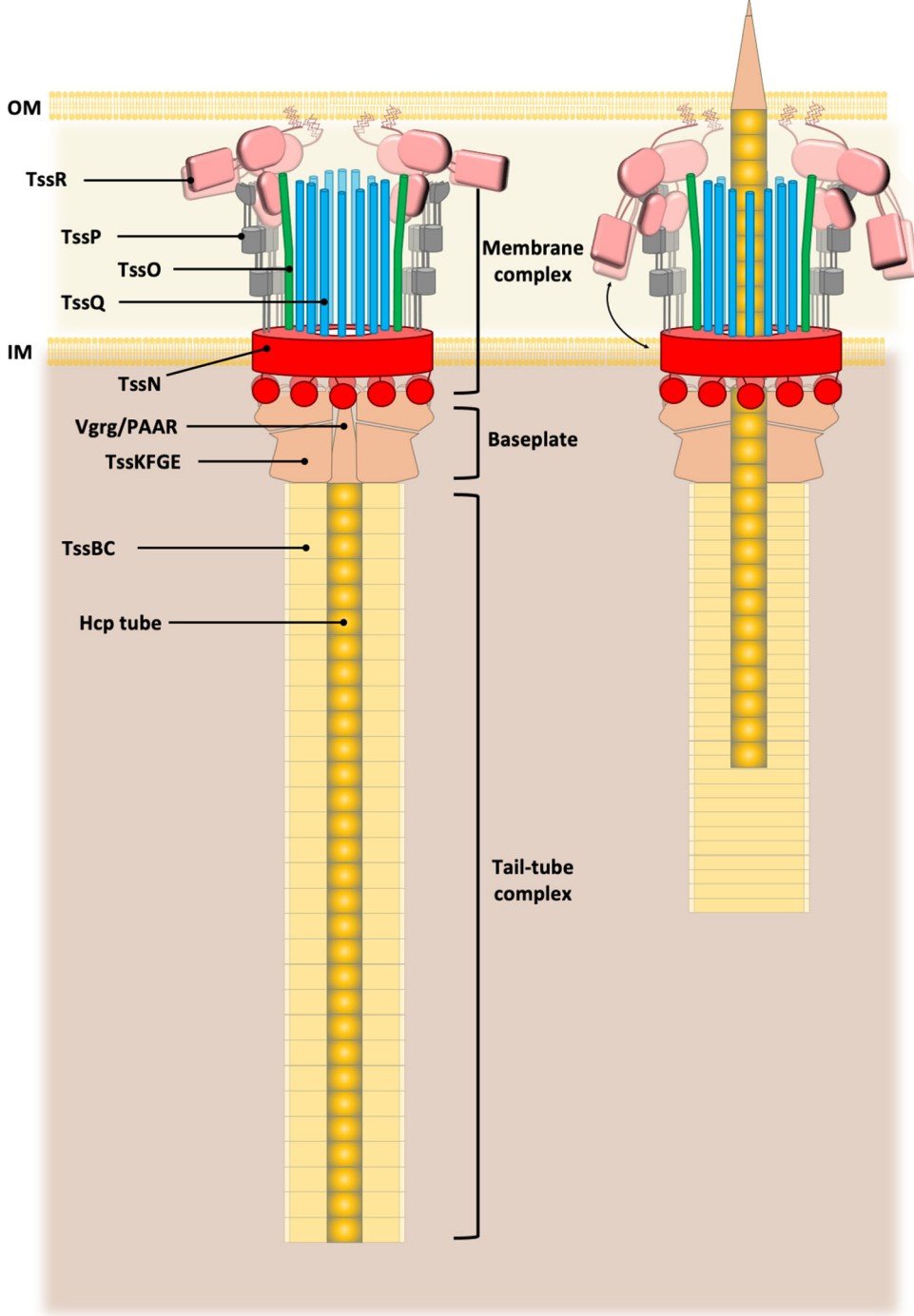

**Fig. 7 | Model of the Bacteroidota T6SS$^{iii}$ assembly.** The T6SS$^{iii}$ is a contractive nano weapon composed of three subcomplexes, the membrane complex, the baseplate and the contractive tail. The newly discovered membrane complex is made by TssN-TssQ-TssO-TssP-TssR and anchor the whole machinery to the membranes of the bacteria. The baseplate and contractive tail are made of TssK-TssF-TssG-TssE and TssB-TssC-Hcp respectively.

## Plasmid construction

PCR was performed using Q5 Polymerase (New England Biolabs). Restriction enzymes were purchased from New England Biolabs and used according to manufacturer instructions. Custom oligonucleotides were synthesized by IDT and are listed in Supplementary Table 2. *Bacteroides fragilis* NCTC 9343 genomic DNA was used as a template for all PCRs. Construction of all plasmids was performed by restriction cloning or Gibson Assembly. Briefly, the genes of interest were PCR-amplified using primers listed in Supplementary Table 2. For plasmids used in protein expression, primers introduced a C-terminal or N-terminal epitope tag extension and restriction sites. pRSF TssQ-GST were constructed by restriction-free cloning: the gene of interest was amplified with oligonucleotides carrying 5' extensions annealing to the target vector. The product of the first PCR was then used as oligonucleotide for a second PCR using the target vector as a template. For construction of the plasmid for constitutive expression of TssB-sfGFP, we fused *sfGFP* amplified from pWW3452 to *tssB* separated by the 6-codon linker ala-ala-ala-gly-gly-gly, as previously published[27]. We then used Gibson Assembly to insert *tssB-sfGFP* into pNBU2-ermGb-P1T-DP-A21 linearized by restriction digestion with NcoI and SalI[26]. All constructs were verified by DNA sequencing (Genewiz).

## Genetic manipulation

Integration of pNBU2 plasmids into *att* sites was performed through mating between *Escherichia coli* S17-1 λ *pir* containing the specific pNBU2 plasmid (see Supplementary Table 2) as previously described with some modifications[18]. For mating, a volume of 25 mL of *E. coli* was incubated in LB to late exponential phase ($OD_{600}$=0.5) on an orbital shaker, and a volume of 3 mL of BHIS media inoculated with *B. fragilis* was incubated to exponential phase under anaerobic conditions. Each culture was mixed, pelleted, and washed before plating at high density on non-selective TSA blood agar media and incubated for 16h at 37°C in an aerobic incubator. The cells were resuspended in 1 mL of BHIS broth and plated on BHIS agar plates containing gentamicin and either erythromycin or tetracycline for *B. fragilis*. The insertions were verified by PCR. Att1 insertions were used in all cases. For the generation of in-frame chromosomal deletions using pExchange-*tdk* plasmid and its derivatives, mating was performed as described above using E. coli S17-1 l *pir* as the conjugal donor strain. After mating, merodiploid integrants were selected on BHIS agar plates supplemented with erythromycin, streaked to BHIS non-selective agar media, then resuspended and plated on BHIS agar media supplemented with FudR for counterselection. The different genetic mutations were confirmed by gene-specific PCR using a combination of flanking and gene-internal primers.

## Complex production and purification

Plasmid combinations were transformed into the E. coli BL21(DE3) expression strain (Invitrogen). Cells were grown at 37 °C in lysogeny broth (LB) to an A600nm of 0.8, and the expression of the genes was induced with 1.0 mM IPTG for 16 hours at 16 °C. Cell pellets were resuspended in ice-cold 50 mM HEPES pH 8.0, 50 mM NaCl, 1 mM EDTA, supplemented with 100 mg/mL of DNase I, 100 mg/mL of lysozyme, EDTA-free protease inhibitor (Roche), and $MgCl_2$ to a final concentration of 15 mM. The cell suspension was then broken using an Emulsiflex-C5 (Avestin). The broken cell suspension was clarified by centrifugation at 16,000 $g$ for 20 minutes. The membrane fraction was then collected by centrifugation at 100,000g for 45 minutes. Membranes were mechanically homogenized and solubilized in 50 mM HEPES pH 8.0, 50 mM NaCl, 1 mM EDTA, and 3% (w/v) n-dodecyl-β-D-maltopyranoside (DDM, Anatrace) at 4 °C for 16 hours. The suspension was clarified by centrifugation at 100,000 $g$ for 45 minutes. The supernatant was loaded onto a 1-mL StrepTrap HP (GE Healthcare) column for Strep tags and then washed with 50 mM HEPES pH 8.0, 50 mM NaCl, and 0.05% (w/v) DDM (affinity buffer) at 4 °C. The complex was eluted in the affinity buffer supplemented with 2.5 mM desthiobiotin (IBA). 1-mL HisTrap HP (GE Healthcare) columns were used for 8-histidine tags and eluted with 250 mM imidazole (Thermo Scientific Chemicals). 5-mL Protino GST (Macherey Nagel) columns were used for GST tags and eluted with 10 mM L-glutathione reduced (Sigma).

## Gel filtration and mild crosslinking

Peak fractions were pooled and loaded onto a Superose 6 10/300 column (GE Healthcare) equilibrated in 50 mM HEPES pH 8.0, 50 mM NaCl, and 0.025% (w/v) DDM. The column was previously loaded with 0.25% glutaraldehyde and run for 5 mL. The TssNQOPR complex eluted as a single monodispersed peak close to the void volume of the column. The sample was used immediately for EM sample preparation.

## Electron microscopy (EM) and image processing

Observation of the TssNQOPR membrane complex or TssN multimers was achieved by negative-stain EM. Five microlitres of the purified complex sample were spotted to glow-discharged carbon-coated copper grids (Agar Scientific). After 30 s of absorption, the sample was blotted, washed with three drops of water and then stained with 2% uranyl acetate. Images were recorded automatically using Gatan

Latitude S software on an TECNAI microscope operating at a voltage of 200 kV and a defocus range of 0.6–25 nm, using an FEI Falcon-II detector (Gatan) at a nominal magnification of 50,000, yielding a pixel size of 2.1A°. A dose rate of 30 electrons per square angstrom per second, and an exposure time of 1 s, were used. A total of 21598 particles were automatically selected from 663 independent images and extracted within boxes of 300 pixels using CryoSPARC[44]. The 2D classification was performed using CryoSPARC.

## Cross-linking Mass Spectrometry

The same sample used for EM grid preparation was also used for XL-MS. $DSS_{H12/D12}$ (Creative Molecules) was added to 1mM final concentration and the sample was incubated for 30 min at 37°C with 600rpm shaking. Reaction quenching was performed adding Tris/HCl to a 20mM final concentration. Samples were processed essentially as described (CITE) In short, samples were reduced, alkylated and digested with 0.5 μg/μL trypsin (Promega). Digested peptides were separated from the solution and retained by a solid phase extraction system (SepPak, Waters), and then separated by size exclusion chromatography to enrich for crosslinked peptides. To remove any remaining interfering solutions a HiPPR™-detergent removal kit (Thermo Scientific™) was used. Liquid chromatography (LC)-MS/MS analysis was performed on an Orbitrap Fusion Tribrid mass spectrometer (Thermo Scientific™). Samples were measured in technical duplicates. Mass spectrometry data was acquired in data-dependent acquisition mode with a cycle time of 3s. Full MS scans were obtained at resolution of 120 000 and scan range of 400-1500 m/z, normalized automated gain control target of 50% and a maximum injection time of 50ms and quadrupole isolation using a 1.6 m/z window. Precursor selection was obtained by monoisotopic peak determination set to peptides with an intensity threshold of 5e3 and charge states of 3-8. Selected peptides were fragmented by collision induced dissociation (CID) with 35% activation energy and dynamic exclusion duration of 60s and MS2 scans were obtained by an Ion Trap detector using rapid scan rate. Acquired data was searched using xQuest in ion-tag mode and cross-links were filtered (deltaS < 0.95, ld score ≥ 25) to a final FDR < 5% using xProphet. Models were predicted using AlphaFold2.

## SDS−PAGE, protein transfer, immunostaining and antibodies

SDS−PAGE was performed on Bio-Rad Mini-PROTEAN systems using standard protocols. For immunostaining, proteins were transferred onto 0.2-μm nitrocellulose membranes (Amersham Protran). Immunoblots were probed with primary antibodies and goat secondary antibodies coupled to alkaline phosphatase, and developed in alkaline buffer in the presence of 5-bromo-4-chloro-3-indolylphosphate and nitro-blue tetrazolium. The anti-HA (HA-7 clone, Sigma Aldrich), anti-FLAG (M2 clone, Sigma Aldrich), anti-StrepII (Sigma Aldrich), anti VSV-G (Sigma Aldrich) and anti-5His (Sigma Aldrich) monoclonal antibodies, and mouse secondary antibodies (Millipore) were purchased as indicated.

## Fluorescence microscopy and analysis

Microscopy was performed on *B. fragilis* cells collected from 48 hr 37 °C anaerobically grown plates exposed to room air. Equivalent populations of cells from whole plate scrapings as determined by $OD_{600}$ were washed twice in PBS with 3 min 8000rcf spins before final resuspension in PBS. Equal volumes of cell suspensions were mounted on 1% agarose pads covered by glass coverslips with the corners stabilized by drops of valap sealant (equal parts petroleum jelly, lanolin, and paraffin). Images were obtained using a Nikon Eclipse 90i inverted fluorescence microscope, with a 60x plan apo oil objective 1.4 NA. Images represent Z max projections of 0.3 μm steps processed using ImageJ. Micrographs presented in the figures display the same signal intensity thresholds for GFP fluorescence and phase contrast across all strains in a given experiment.

The MicrobeJ ImageJ plugin was used for quantifying bacteria and maxima[45]. Bacteria were counted using the following criteria applied to phase contrast channels: medial axis, 0.4-1.6 area, 0.65-2.7 length, 0.5-1.3 width, 0-0.94 circularity, 0-0.25 curvature, 0-0.2 angularity. Maxima settings varied by experiment due to differences in background fluorescence, thus each experiment's parameters were set by comparing positive control (wildtype) and a baseplate-deficient negative control (ΔtssK). Across all replicates, the following maxima settings were used: foci, basic, 2000 tolerance, 0.07-1.25 area, 0-1.25 length, 0-0.7 width. Replicate 1 used 3.5 Z score and 16000-max intensity, replicate 2 used 11 Z score, 14000-max intensity, replicate 3 used 5 Z score, 14000-max intensity, replicate 4 used 3.5 Z score, 10500-max intensity. Heatmaps were generated using MicrobeJ XYCellDensity plots performed on ~500 randomly-selected maxima across all strains to control for varying foci densities. Percent polar localization was determined using the MicrobeJ classifications of "polar," "midcell," and "betwixt" for maxima localization in parent bacteria.

### T6SS competition assay
Competitions were performed on strains co-incubated under anaerobic conditions at 37 °C for 16 hr on blood agar plates (A10, Hardy Diagnostics). Initial suspensions were mixed in a 10:1 $OD_{600}$ ratio of donor (*B. fragilis* NCTC 9343) to recipient cells (*B. thetaiotaomicron* VPI-5482). Colony forming units (CFU) were enumerated on selective agar plates (6 µgmL-1 tetracycline for recipient, 12.5 µgmL-1 erythromycin for donor) for initial suspensions and final suspensions. The competitive indices were calculated by dividing the final donor:recipient ratio by the initial donor:recipient ratio. Each competition was performed in technical triplicate before being averaged as a single experimental replicate. For complementation strains, the deleted gene was restored with an ectopic chromosomally-inserted plasmid at the att1 insertion site.

### T6SS Hcp secretion ELISA
*B. fragilis* strains were streaked on BHIS agar medium supplemented with 1 ugml-1 of vitamin K3 (Acros Organics/Fisher Scientific) and gentamicin (60 ugml-1). Plates were incubated for two days under anaerobic conditions. Nine clones per strain were inoculated in 4 ml of BHIS and incubated for 16 hr in anaerobic conditions. A volume of 500 ul of the cultures was diluted in 5 ml of fresh BHIS, and the cultures were incubated under anaerobic conditions for 3–4 hr. The $OD_{600}$ of each culture was measured. For each culture, 1 ml was transferred to an Eppendorf tube. The Eppendorf tubes were centrifuged at 3000 x *g* for 15 min. A volume of 500 ul of supernatant was then transferred to a new eppendorf tube. The supernatant was centrifuged again in the same conditions, and 75 ul was added to a Stripwell Microplate flat bottom ELISA with high binding (Corning). The ELISA was performed as described previously with some modifications[18]. For the binding step, the plate was incubated 16 hr at 4 °C on a rocking platform. For the blocking step, incubation was performed for 1.5 hr. For the primary antibody incubation step, the anti-TssD (Hcp) rabbit primary was a gift from Harris D. Bernstein, used at a 1:2000 dilution, and incubated for 2 hr[46]. The antibodies HRP Donkey anti-rabbit IgG (Biolegend) were used at a 1:7500 dilution. For quantification, TBM one solution (Promega) was used. The reaction was stopped using 50 ul of 2M sulfuric acid and $OD_{450}$ was measured, normalized by the $OD_{600}$ of the starting culture. The experiment was repeated twice. Uninoculated media was used as a blank. For the ΔtssP condition, there were only 8 replicates.

### TssNc overexpression experiments
*B. fragilis* TssNc strains harbor wildtype *tssN* or in-frame chromosomal deletions of *tssN*, and chromosomal single copy insertions of *tssNc* under constitutive highly expressing promoter (P1T$_{DP}$[A21]) or constitutive moderately expressing promoter (BT1311)[26]. TssNc was defined as the sequence from amino acid 156 to the end of the gene (stop codon at 283). TssNc was generated as a gene block (gBlock) from IDT with the following following homology flanks to pNBU2: CTCCAAATCTGTTTTTTAACA and AACTAGTGGATCCCCCGGGC. The gBlock was inserted via Gibson Assembly into pNBU2_ErmGB_BT1311 linearized by restriction digestion with NdeI and XbaI and transformed into *E. coli* S17-1. For the high expression promoter, *TssNc* was amplified using the following primers: 5-TTTATGATATTAAACGAATCATGCCGATTCCCGTGTATG and 5-GTAATGGAACATAATGAGAACTAACTAATTGCCTATCTTCCAG. The amplicon was gibson ligated into pNBU2_Erm_P1T$_{DP}$[A21] restriction digested with NcoI and SalI, and transformed as before. S17-1 strains harboring TssNc plasmids were mated into *B. fragilis* wildtype or *tssN* at att1 insertion sites.

### De novo modelisation, domain prediction, and model validation
TssN, TssP, TssO, TssQ, and TssR protein models were built using AlphaFold2 on a local Ubuntu installation with the "monomer_ptm" preset and "full_dbs"[22]. Of the five models generated, we chose to analyze and display the top-ranked model. Protein-domains were predicted using the SWORD2 web server[47]. To validate the predicted AlphaFold2 models, the identified crosslinks with ld score>25 were mapped on the structures of the five proteins and classified into satisfied or violated using a threshold distance of 35 Ang.

### Mass Spectrometry for protein identification
The list of identified proteins provided in Fig. 2a was established from mass spectrometry–based proteomics, on excised SDSPAGE gel bands (duplicates) containing the elution of the *B. fragilis* pulldown. Slight modifications were brought: After in-gel trypsin digestion and LC-MSMS analysis on an Ultimate 3000 liquid chromatography coupled to a Q-Exactive plus mass spectrometer (ThermoFisher), spectra were processed by Proteome Discoverer software (ThermoFisher, version 2.4.1.15) using the Sequest HT algorithm with the databases extracted from Uniprot, *Bacteroides fragilis* NCTC 9343. The estimated stoichiometry was calculated with the emPAI quantification with the following formula:

$$Estimated\,stoichiometry = 2 \times emPAI = 2 \times 10^{\frac{N_{observed}}{N_{observable}}} - 1$$

### Genome analysis
The data for 1253 completely sequenced genomes were retrieved from NCBI RefSeq (see accession number in Source Data Files 1–4, last accessed March 2023) using ncbi datasets CLI tools (version 14.7.0, CLI : datasets download genome taxon bacteroidota --assembly-level complete [Source dataset 2]). The subset of species reference genomes was produced (344 complete genomes, one per species [Source dataset 1]). We also downloaded a third dataset of the reference 1911 genomes including one genome per species, but including genomes that are not assembled to completion [Source dataset 3]. Since an incomplete assembly can affect our conclusions on genetic organization of the systems, this dataset was only used to make the HMM profiles and will not be further mentioned. We pre-processed the genomes with the *prepare* module of PanACoTA (version 1.4.0,[48]), and made a quality control check by selecting the genomes with L90 <= 100 and number of contigs < 999. These genomes were then annotated with prokka (version 1.14.5,[49]) with the annotate module of PanACoTA.

To analyze the genetic organization of the T6SS of Proteobacteria and to check the absence of the T6SS[iii] in other clades we used an additional dataset. We retrieved all the complete genomes of the NCBI non-redundant RefSeq database (ftp://ftp.ncbi.nlm.nih.gov/genomes/refseq/, last accessed in March 2021), including 21084 bacteria. We used the original annotations. We used the taxonomical classification of GTDB (except for the Orders, where we used the NCBI classification).

## Identification of T6SS

The universal components of the T6SS (TssB, TtsC, TssD, TssE, TssF, TssG, TssH, TssI, TssK) and those of the MC (TssN, TtsO, TssP, TssQ, TssR) were searched using hidden Markov models (HMM) with hmmer (version 3.2.1, using GA thresholds for all but TssO and TssQ where a default MacSyFinder e-value of 0.1 was used, knowing that hits are in relevant clusters (version 20230402.dev,[50]). Most of the models used were the ones of TXSScan (version 1.1.1,[30]), but we made changes and updates in some cases (see below). These models include information on the components of the T6SS, their quorum, and their genetic organization. Given that we now have many more genomes of Bacteroidota, we searched to improve the previous model of T6SSⁱⁱⁱ. Notably, we explored the effects of changing the number of required elements for the system and we allowed for systems that were encoded in scattered loci. We also produced more informative profiles for TssO and TssQ. Profiles were built using blastP with the proteins of reference against Source datasets 2 & 3, only significant hits (e-value ≤ 0.05) were retrieved and aligned with MAFFT (version 7.407, options --localpair and --maxiterate 1000,[51]). The HMMs were then built from the multiple sequence alignments using hmmbuild from HMMER (version 3.2.1) after manual trimming of the edges when pertinent. These analyzes led to an improved T6SSⁱⁱⁱ model.

To assess if we needed to make the TXSScan models for T6SSⁱⁱⁱ more permissive, we compared the previously published model for the T6SSⁱⁱⁱ with variants where we decreased the number of required components. We then tested the need to class some components as loners (*i.e.* genes that in some case may be encoded far from the main locus). The differences between the models were inspected to check if the novel systems were complete enough (presence of key markers, absence of obvious pseudogenes). We ended up using a model of T6SS including all the components above, a quorum of 1 minimum mandatory genes where TssN, TssO, TssP, TssQ, TssR were set as mandatory genes, and 8 minimum genes required overall (including the universal components of the T6SS). We activated the multi-loci option because in some genomes the system is encoded in multiple loci (see Results). To verify that MC was systematically associated with the presence of a T6SS, we built a model to search only for universal components in (the MC components were allowed to be absent). This model includes all the universal components as mandatory genes, a minimum of 6 mandatory genes (also the minimum genes required) and we activated the multi loci option. For both models, the inter gene maximum distance was set to 10. All models and profiles are available as supplementary material (Source Data Files 1-4) and will be made available on the git of TXSScan upon publication.

## Analysis of genetic organization

We analyzed the order of the genes encoding the different components of all the T6SSⁱⁱⁱ identified in the Bacteroidota genomes and all the T6SS identified in Proteobacteria. We defined two genes as neighboring if they were at less than 10 genes apart and if none of the intervening genes was a gene encoding one of the other key T6SS components. We then took all pairs of neighboring genes and built two graphs to represent the data. In this graph, the nodes are genes encoding the different components and the edges represent relations of neighborhood. The weight of the edge between two components is proportional to the number of times the two components were found to be neighbors. We represented the data in Gephi (version 0.10.1), using the method Circular Layout and Expansion to optimize the display of the neighborhood network.

## Phylogenetic analyzes

We built two phylogenetic trees of the phylum, one including one taxa per species (taxonomy as defined in RefSeq) and another representing all genomes (Source Datasets 1 and 2). We started by identifying homologs using a set of HMM profiles that match proteins that are ubiquitous and tend to be encoded by one single gene copy in the genomes of Bacteria (Source Data Files 1-4). The analysis of the hits of these HMM profiles revealed that a few were rarely matching in the phylum whereas others have frequently more than one good hit. We therefore restricted the analysis to 128 profiles that were present in more than half of the genomes and typically having a best hit that was much higher than the others. We searched for these proteins in the genomes using hmmer (version 3.3.2, options --tblout, --cpu 20, --cut_ga), and each time took the best hit that had a significant gathering threshold score. The hits for each protein family were aligned using MAFFT (7.505, options --localpair and --maxiterate 1000). The resulting multiple alignments were analyzed with ClipKIT (1.3.0, option --kpi-gappy,[52]) to remove uninformative positions. The curated multiple alignments were then concatenated and given to IQ-tree (2.2.2.2, -st AA -m TEST, -B 1000, -alrt 1000, -T AUTO, --threads-max 28, --mem 100GB,[53]) to obtain a phylogenetic tree by maximum likelihood with the best model being chosen by the program (Q.yeast+F+G4 [Source dataset 1, one genome per species], Q.yeast+G4 [Source dataset 2, all genomes]). The robustness of the phylogenetic trees was assessed using 1000 ultra-fast bootstraps[54]. The data on the presence/absence of the components of the T6SS was plotted in the tree using iTOL[55]. All trees were rooted using as an outgroup the genera *Rodothermus* and *Salinibacter*.

## Reporting summary

Further information on research design is available in the Nature Portfolio Reporting Summary linked to this article.

## Data availability

All data supporting the findings of this study are available within the manuscript and its associated supplementary information. Source data are provided with this paper. The MS raw files have been deposited to the ProteomeXchange Consortium via the PRIDE partner repository with the project accession number PXD042118[56]. Source data are provided with this paper.

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

## Acknowledgements
We thank members of the Durand laboratory and Vladimir Pelicic for discussions, Moly Ba, Isabelle Bringer, and Annick Brun for technical assistance, Justin Sonnenburg for sharing plasmid pWW3452, and Joseph Mougous for support and introductions. This work was funded by the Centre National de la Recherche Scientifique, the Aix-Marseille Université, the Institut National de la Santé Et de la Recherche Médicale and grant from the Agence Nationale de la Recherche (ANR-22-CE11-0005-01) to ED, and National Institutes of Health grants K99GM129874, R00GM129874, and R35GM142685, and startup funds from the Dartmouth College Geisel School of Medicine to BDR. We thank the support of the Institut de Microbiologie de la Méditerranée (IMM) platforms: Proteomics, Microscopy and Protein production facilities. This work used the computational and storage services (TARS cluster) provided by the IT department at Institut Pasteur, Paris. Laboratoire d'Excellence IBEID Integrative Biology of Emerging Infectious Diseases [ANR-10-LABX-62-IBEID], Equipe FRM (Fondation pour la Recherche Médicale, EQU201903007835) to EPCR. FS is grateful for funding from the German Research Foundation (project number 496470458 and CRC969). Microscopy experiments were supported by the Dartmouth College bioMT Core facility through NIH NIGMS grant P20-GM113132.

## Author contributions
E.D. and B.D.R. conceived the project. E.D., T.R.B. and M.V. performed all the heterologous and in vitro study in *E. coli* and pulldown coupled to mass-spectrometry analysis from *B. fragilis* samples. T.R.B. performed all the biochemistry, biophysics and electron microscopy experiments. C.J.L., E.T., S.R., K.S., D.S. and B.D.R. performed all the studies in *B. fragilis* with the help of MV for construction of the allelic exchange plasmids. A.K. helped with electron microscopy and data collection. J.O and F.S. performed the crosslinking mass spectrometry (CXMS) experiments with the help of RP for data analysis. RP helped with AlphaFold2 models and their validation with CXMS data. Y.L.C. and E.P.C.R. performed all the phylogenetics studies. E.D. and B.D.R. wrote the manuscript with contribution from all the authors. All authors contributed to the article and approved the submitted version.

## Competing interests
The authors declare no competing interests.
