## [Peer Review File - NEW · Nature Communications]

Assembly of a unique membrane complex in type VI secretion systems of BacteroidotaReviewer #1 (Remarks to the Author):

In this manuscript Bongiovanni and colleagues describe a preliminary composition of the membrane complex of type 6 secretion system sub-type T6SSiii, specific to Bacteroidota phylum. They provide structure predictions of protein components, measure stoichiometry, interaction of individual components by pull-down assays and provide comprehensive phylogenetic overview. Another important result is the interaction of TssN cytoplasmic domain with a baseplate protein TssK. While overall it is novel and important to the field and I am convinced about the composition of the *Bacteroides fragilis* membrane complex, I have remarks about most of the experimental data in this manuscript.

1. EM images (Figure 2, 4, Extended Data F5) seem to be highly heterogenous. It is difficult to appreciate the similarity of the objects boxed in Figure 2E and one can see many different sized particles in the background. Although, methods say that over 20,000 particles have been picked, the 2D classes are very poor quality in the Extended Figure 5.
2. Fluorescent microscopy images (Figure 5) are very poor quality. It is difficult to appreciate what the authors call "foci".
3. No protein gels are shown, only highly cropped western blots, therefore it is very difficult to estimate quality/purity of purifications.
4. While other baits for pull-down assays were smaller tags such as strep or his, TssQ was GST-tagged. GST measures over 20 kDa and I think it oligomerizes, so it is overall poor choice to study interactions.
5. Authors do not comment on inter-protein interactions in their Cross-linking experiments with TssNQOPR complex. Were there any inter-protein cross-links? Do they support pull-down assays and schematic representation in Figure 3?
6. The order of calling figures in the manuscript is not consecutive. Figure 2 c is called first, then a and b. Extended Data Figure 4 is called before 2 and 3.
7. TssN, TssP, O, Q have already been predicted to have transmembrane segments and with TssR to constitute the membrane complex in Bacteroidota (Russel et al 2014 Cell Host Mic and Coyne et al., BMC Genomics 2016). These references are used (Line 96), but the following sentence says "It is unknown how the T6SS could function without an MC", which ignores the previous hypotheses in the literature and overall sounds like a very naïve hypothesis. This rhetorical question is raised in the abstract as well, and I think it should be replaced by a parsimonious hypothesis that is actually pursued in this manuscript, i.e. that an alternative complex allowing passage of T6SS needle should exist in T6SSiii.
8. Figures 1, 5 and 7 all have different graphical representation of T6SS components. Authors should make effort to keep the same style.

Minor remarks:

Line 80 – to be precise, VgrG is trimeric, but PAAR is monomeric.

Line 87 – should be more precise, what type of phages.

Line 100 – should be "what are their patterns".

Lines 125-130– word use for EccD is a "match", while for TssQ-O are "distant structural similarity", although Z scores and rmsd are similar.

Line 157 – at this point "membrane association" is still a prediction.

Line 200 – should be better explained what authors mean by "violated", it sounds strange.

Reviewer #2 (Remarks to the Author):

The authors have analyzed the T6SS found in *B. fragilis*, which is distinct from the canonical T6SSs of other bacterial species. Notably, they have identified the membrane complex anchoring the *B. frag* T6SS to the membrane and performed basic structural prediction, protein-protein interaction, and stoichiometric analysis of this complex.

Overall, the science is done about as well as can be done. Short of reconstituting and obtaining the structure of the entire membrane complex, the authors have done about all they can. Protein-

protein interactions presented are based solely on pairwise pulldowns – some secondary confirmation of these interactions or modeling of how these domains might fit together (the model figure presented doesn't really take into account the stoichiometry worked out in their study) would be nice, but maybe that's not feasible.

My only gripes (all relatively minor) are with how the data is being presented:

Maybe I missed it, but how reproducible were the pulldown experiments? I see the statement about the competition experiments having lots of replicates, but I didn't see a similar statement about the pulldowns.

For figure 5, the percent of cells with foci is presented in a very odd way... normalizing the number of foci in this way feels a bit sketchy. I would have thought that the immunostaining of WT and each mutant would be independently stained – it's not like all strains are being stained on the same slide together, right?. Wouldn't this data be best presented as raw % of cells with foci across all mutants?

Aside from the fact that immunostaining microscopy of bacteria is prone to artifacts (some of the foci in the representative micrographs don't really look like foci), I didn't see any statement of how many cells were being analyzed/counted to generate these numbers. How many cells are in the "four microscopy fields" for each point. Considering the data represented are ~10% of a 5% normalized 100, that means that the relative frequency of foci is something like 0.5%, so if each microscopy field isn't several thousand cells, I'd be a bit worried about potential bias. Maybe more details about the precise MicrobeJ settings that were used to call "intense IF staining" as "foci"?

What is the difference between the red and green traces in figure 4B?

Reviewer #3 (Remarks to the Author):

This manuscript by Bongiovanni et al is focused on the *Bacteroides fragilis* T6SS. In Proteobacterial T6SS machines there is a minimum of 13 essential proteins which make up 3 sub-complexes namely, the tail-tube complex, baseplate and the membrane complex, the latter of which provides a channel between the inner and outer membranes. While homologs of the tail-tube and baseplate have been identified in Bacteroidales T6SS, there have not been any homologs of the membrane complex identified, which poses the question of how the T6SS in these bacteria is able to function. In the current study the authors identify and characterise the components in the membrane complex of the T6SS in *B. fragilis*, which is composed of 5 different proteins, TssNQPOR. The authors also present the solved structure of the TssNQPOR complex as well as biochemical data on how these components likely interact with one another in vivo. Additionally, the authors present data from phylogenetic analyses on the distribution of these newly identified membrane complex genes across the phylum Bacteroidota and postulate as to how these bacteria may have co-opted these genes as functional components of their T6SS machinery.

Overall, I found the manuscript to be very interesting and well written, and to nicely address a key aspect of T6SS biology in Bacteroidales. I think it will be of interest to researchers in the T6SS and general secretion system fields. I have a couple of major comments that should be addressed in a revised manuscript and several minor comments, as outlined below.

Major comments:

Lines 163-164: Did the authors confirm at any point that the STREP and HIS tags on TssN and TssO at the native location did not impact T6SS function? The authors should use their T6SS competition assay and the Hcp secretion ELISA assay to confirm that the tagged versions of these proteins have wildtype T6SS functionality.

Lines 271-276 and Figure 5a-c: The authors should show representative fluorescent images of the TssK foci that were used to determine the number of foci/total cells in Figure 5a as has been done for Fig 5b with the images in Fig 5c. In both cases it would be better to include a larger region of the field of view for this than just one or two cells – the authors claim that the images in Fig 5c are representative of the data shown in Fig 5b but this doesn't make sense as each mutant strain has no TssK foci in any of the cells in Fig 5c but in the graph in Fig 5b there are still some foci detected in all mutant backgrounds.

The authors should also provide a citation(s) for the statement in line 273 "...consistent with previously reported frequencies in other bacteria."

Lines 277-279 and Figure 5d-e: The same is true in Fig 5d-e as for the above point in Figs 5b-c: the apparently representative images in Fig 5e do not line up with the number of foci/cells reported in 5d i.e. there are still some foci in the low expression condition but no foci for this same condition in Fig 5e. As above, this should be addressed by including larger fields of view from the microscopy image that has more than just 2 cells per field.

Minor comments:

Line 157: the authors claim at this point to have identified "five membrane associated proteins". At this point in the manuscript the data only demonstrates that these proteins are essential for T6SS function and that they are likely membrane associated from in silico analyses but not yet demonstrated to be membrane associated. I would suggest to revise to say "five putative membrane associated proteins".

Figure 2a: It would be helpful to define what the abbreviations are for each column title in the figure legend to help the reader understand the information in the table.

Figure 3 and lines 861-863: the authors are proposing elsewhere in the manuscript that the whole complex is anchored to the outer membrane by putative lipoprotein TssR – in this figure there is a pink arm extending horizontally off TssR but it might make more sense if this arm is actual vertical and contacting the outer membrane in the figure? Also the figure legend for this says that "The full arrows represent copurification observed in full length proteins, the dashed arrow represent copurification of the periplasmic truncations of the proteins." However, the lines in the actual figure are black with arrow heads or grey with blunt ends – this should be corrected in the figure/legend so both match.

Figures 5c and 5e: the left-hand image in both cases is the composite of the TssK fluorescent image and the phase contrast image – it is really hard to actually see the TssK fluorescent foci in these composite images over the background fluorescence in the whole cell, compared to the much clearer fluorescent foci in the middle images. Why is this? Have the authors altered the brightness and contrast or maximum levels of this fluorescent channel differently in the left-hand composite image compared to what is presented in the middle image? I would suggest correcting this so the foci are clear in the composite image, or alternatively remove the composite images as it is quite confusing.

Line 442: "...found in protein..." should be "...found in proteins..."

REVIEWER COMMENTS

Reviewer #1 (Remarks to the Author):

In this manuscript Bongiovanni and colleagues describe a preliminary composition of the membrane complex of type 6 secretion system sub-type T6SSⁱⁱⁱ, specific to Bacteroidota phylum. They provide structure predictions of protein components, measure stoichiometry, interaction of individual components by pull-down assays and provide comprehensive phylogenetic overview. Another important result is the interaction of TssN cytoplasmic domain with a baseplate protein TssK. While overall it is novel and important to the field and I am convinced about the composition of the *Bacteroides fragilis* membrane complex, I have remarks about most of the experimental data in this manuscript.

We thank the reviewer for their assessment of our manuscript and address their comments in detail below.

1. EM images (Figure 2, 4, Extended Data F5) seem to be highly heterogenous. It is difficult to appreciate the similarity of the objects boxed in Figure 2E and one can see many different sized particles in the background. Although, methods say that over 20,000 particles have been picked, the 2D classes are very poor quality in the Extended Figure 5.

We thank reviewer #1 for the comment and we are aware of this heterogeneity of the purified TssNQOPR complex. Actually, this problem has prevented us from (1) getting nice and sharp 2D classes and (2) obtaining the high resolution 3D reconstruction of the whole complex. Nevertheless, so far it is the best we can achieve with this type of sample. Future work will be dedicated to improve the biochemistry towards the goal of solving the high-resolution structure of the T6SSⁱⁱⁱ MC. So right now it is slightly beyond the scope of the current work.

2. Fluorescent microscopy images (Figure 5) are very poor quality. It is difficult to appreciate what the authors call “foci”.

All reviewers had concerns with the quality of the anti-TssK fluorescence microscopy data in the original submission. We tried hard to improve this aspect of our study but were unable to make significant improvements. Therefore, we have replaced the TssK data with entirely new data using a strain of *B. fragilis* that constitutively overexpresses a sheath protein fused to sfGFP (TssB-sfGFP), recently demonstrated by the Comstock and Basler groups to yield robust detection of stereotypical T6SS sheath dynamics in live cells under conditions of very low oxygen (Garcia-Bayona *et al* 2021 PNAS). We engineered WT and MC mutant strains to produce TssB-sfGFP and repeated all of the analysis from the original submission. Notably, our major findings from the TssK data hold in the new analysis: MC mutants exhibit significantly less sheath foci than does the wildtype T6SS-functional strain. We also believe that the image quality of the new data is much “nicer” and this allowed an opportunity to collect data from a higher number of cells, which we then leveraged to examine subcellular localization of sheath foci in each strain using heatmaps (see Figure 5).

3. No protein gels are shown, only highly cropped western blots, therefore it is very difficult to estimate quality/purity of purifications.

The uncropped data are also presented in the extended data. For clarity we choose to present only the cropped version in the main figure to highlight the specific band corresponding to the protein of interest. We would like to raise the attention that all these

protein gels are western blots, not Coomassie staining. So they do not intend to estimate the purity but only the presence or absence of the protein.

4. While other baits for pull-down assays were smaller tags such as strep or his, TssQ was GST-tagged. GST measures over 20 kDa and I think it oligomerizes, so it is overall poor choice to study interactions.

We thank reviewer #1 for the comment and we have purified TssQ^{GST} + TssO^{HIS} complex and show by calibrated gel filtration that it is monomeric (1 TssQ^{GST} + 1 TssO^{HIS}). Please see the additional data here:

We would point out these additional observations that strengthen this point:

1. TssN-STREP does interact with TssQ-FLAG: so TssQ is capable of interacting with another protein even without a GST tag.
2. TssQ-GST does not interact with TssK: so GST on TssQ does not trigger unspecific interaction with any protein

5. Authors do not comment on inter-protein interactions in their Cross-linking experiments with TssNQOPR complex. Were there any inter-protein cross-links? Do they support pull-down assays and schematic representation in Figure 3?

The reviewer is correct in assuming that we had also identified multiple inter-protein cross-links. These inter-protein links were overall in a good agreement with our other data. We did however also identify a number of connections within the TssNQOPR complex, which we currently cannot fully rationalize. We at least partly ascribe this to the fact that our enrichments still contain a certain amount of polydispersity, as can also be seen from the EM data. We therefore had decided to only use the identified intra-protein crosslinks for validation of the TssNQOPR complex model for this study, as these should not be affected. Nonetheless, we did include all inter-protein crosslinks for the TssNQOPR complex already in our original submission (see "Extended Data File 4") and have also uploaded them to PRIDE.

6. The order of calling figures in the manuscript is not consecutive. Figure 2 c is called first, then a and b. Extended Data Figure 4 is called before 2 and 3.

We thank reviewer #1 for the comment and we apologize. We have taken great care to cite the figures in a logical and consecutive way throughout the revised manuscript. For Figure 2, we swapped panel 2b and 2c to follow the main text. Extended Figure “4” becomes “2”, and “2” becomes “3” and “3” becomes “4”.

7. TssN, TssP, O, Q have already been predicted to have transmembrane segments and with TssR to constitute the membrane complex in Bacteroidota (Russel et al 2014 Cell Host Mic and Coyne et al., BMC Genomics 2016). These references are used (Line 96), but the following sentence says “It is unknown how the T6SS could function without an MC”, which ignores the previous hypotheses in the literature and overall sounds like a very naïve hypothesis. This rhetorical question is raised in the abstract as well, and I think it should be replaced by a parsimonious hypothesis that is actually pursued in this manuscript, i.e. that an alternative complex allowing passage of T6SS needle should exist in T6SSiii.

We thank reviewer #1 for the comment and we have modified the manuscript to incorporate this comment.

We made the following changes to the text:

lines 49-50: “How a T6SS might function without a canonical MC is unknown or do T6SSⁱⁱⁱ use an alternative MC?”

lines 99-100: “Another possibility is that an alternative complex allowing passage of T6SS needle should exist in T6SSⁱⁱⁱ.”

8. Figures 1, 5 and 7 all have different graphical representation of T6SS components. Authors should make effort to keep the same style.

We thank reviewer #1 for the comment and we totally agree. We have reshaped the graphical representation so that they all have the same style.

Minor remarks:

We have made all the changes necessary to address these remarks in the main text.

Line 80 – to be precise, VgrG is trimeric, but PAAR is monomeric.

Line 87 – should be more precise, what type of phages. We added “T4, Mu and Siphophage”.

Line 103 – should be “what are their patterns”.

Lines 129 – word use for EccD is a “match”, while for TssQ-O are “distant structural similarity”, although Z scores and rmsd are similar. We used the same description for both cases: “distant structural similarity”.

Line 162 – at this point “membrane association” is still a prediction. We added “putative” membrane associated proteins.

Line 206-208 – should be better explained what authors mean by “violated”, it sounds strange. “Violated”, in computational structural biology jargon, refers to a distance restraint which is not satisfied. Here the crosslinks detected by Mass Spectrometry are interpreted as distance restraints and, once mapped on the AlphaFold2 models, they might be satisfied or not satisfied (i.e., violated) when the distance of the two corresponding residues exceeds the maximal length of the crosslinker. We thus rephrased the sentence: “We obtained 64 high-confidence unique crosslinks, of which only 12% are violated, for which the distance of the two corresponding residues exceeds the maximal length of the crosslinker, mainly in TssQ (6 violations over a total of 18 crosslinks).”

Reviewer #2 (Remarks to the Author):

The authors have analyzed the T6SS found in *B. fragilis*, which is distinct from the canonical T6SSs of other bacterial species. Notably, they have identified the membrane complex anchoring the *B. frag* T6SS to the membrane and performed basic structural prediction, protein-protein interaction, and stoichiometric analysis of this complex.

Overall, the science is done about as well as can be done. Short of reconstituting and obtaining the structure of the entire membrane complex, the authors have done about all they can.

We thank the reviewer for their positive comments and assessment of our manuscript. We address points raised in the responses below.

Protein-protein interactions presented are based solely on pairwise pulldowns – some secondary confirmation of these interactions or modeling of how these domains might fit together (the model figure presented doesn't really take into account the stoichiometry worked out in their study) would be nice, but maybe that's not feasible.

We thank reviewer #2 for the comment and we did our best to change our final model to better fit with the data (stoichiometry and PPI). Please see Figure 7.

My only gripes (all relatively minor) are with how the data is being presented:

1. Maybe I missed it, but how reproducible were the pulldown experiments? I see the statement about the competition experiments having lots of replicates, but I didn't see a similar statement about the pulldowns.

We added this sentence (lines 222-223): "pulldown experiments have been reproduced several times with similar results". In addition we have improved the presentation of the purification protocol (see lines 572-598).

2. For figure 5, the percent of cells with foci is presented in a very odd way... normalizing the number of foci in this way feels a bit sketchy. I would have thought that the immunostaining of WT and each mutant would be independently stained – it's not like all strains are being stained on the same slide together, right?. Wouldn't this data be best presented as raw % of cells with foci across all mutants?

We appreciate this point. We found that the raw percent of foci in the wildtype strain exhibited substantial variation between biological replicates, ranging across a nearly 10-fold scale (see figure below). As the field still does not fully understand how the GA3 T6SS is regulated, we do not fully understand why this occurs, however, to account for this we decided to perform normalization per biological replicate. It is apparent however that the major findings concur, whether data are presented as raw % or normalized.

3. Aside from the fact that immunostaining microscopy of bacteria is prone to artifacts (some of the foci in the representative micrographs don't really look like foci), I didn't see any statement of how many cells were being analyzed/counted to generate these numbers. How many cells are in the "four microscopy fields" for each point. Considering the data represented are ~10% of a 5% normalized 100, that means that the relative frequency of foci is something like 0.5%, so if each microscopy field isn't several thousand cells, I'd be a bit worried about potential bias. Maybe more details about the precise MicrobeJ settings that were used to call "intense IF staining" as "foci"?

We have now provided extensive detail regarding MicrobeJ settings used in microscopy image analysis in the Materials and Methods, as well as the total numbers of cells and foci enumerated and analyzed in these studies.

4. What is the difference between the red and green traces in figure 4B?

We thank reviewer #2 for the comment. We have modified Fig. 4b for better clarity. The previous DLS graph (Figure 4b) showed the TssN homomultimer elution intensity/size (nm) ratio before ultracentrifugation at 100,000 relative centrifugal forces (rcf) for 20 minutes in red, and after ultracentrifugation in green. The red curve was removed for the revision, showing only the sample used for electron microscopy (Figure 4c).

Reviewer #3 (Remarks to the Author):

This manuscript by Bongiovanni et al is focused on the *Bacteroides fragilis* T6SS. In Proteobacterial T6SS machines there is a minimum of 13 essential proteins which make up 3 sub-complexes namely, the tail-tube complex, baseplate and the membrane complex, the latter of which provides a channel between the inner and outer membranes. While homologs of the tail-tube and baseplate have been identified in Bacteroidales T6SS, there have not been any homologs of the membrane complex identified, which poses the question of how the T6SS in these bacteria is able to function. In the current study the authors identify and characterise the components in the membrane complex of the T6SS in *B. fragilis*, which is composed of 5 different proteins, TssNQOPR. The authors also present the solved structure of the TssNQOPR complex as well as biochemical data on how these components likely interact with one another in vivo. Additionally, the authors present data from phylogenetic analyses on the distribution of these newly identified membrane complex genes across the phylum Bacteroidota and postulate as to how these bacteria may have co-opted these genes as functional components of their T6SS machinery.

Overall, I found the manuscript to be very interesting and well written, and to nicely address a key aspect of T6SS biology in Bacteroidales. I think it will be of interest to researchers in the T6SS and general secretion system fields. I have a couple of major comments that should be addressed in a revised manuscript and several minor comments, as outlined

below.

We thank the reviewer for their overall positive assessment of our manuscript.

Major comments:

Lines 169-170: Did the authors confirm at any point that the STREP and HIS tags on TssN and TssO at the native location did not impact T6SS function? The authors should use their T6SS competition assay and the Hcp secretion ELISA assay to confirm that the tagged versions of these proteins have wildtype T6SS functionality.

The reviewer raises a good point. To address this, we have performed competition experiments in which we assess competitive index of single- and doubly-tagged strains in comparison to the wildtype. We find that each single tag exhibits wildtype T6SS function. Although the double-tagged strain is slightly diminished in function, it still retains ~100fold higher functionality compared to a T6SS-inactive sheath-mutant control strain. We have included these data in the supplemental figures and add the following sentence (line 169): “without altering the functionality of the T6SS (Extended Data Fig. 1b).”

Lines 274-282 and Figure 5a-c: The authors should show representative fluorescent images of the TssK foci that were used to determine the number of foci/total cells in Figure 5a as has been done for Fig 5b with the images in Fig 5c. In both cases it would be better to include a larger region of the field of view for this than just one or two cells – the authors claim that the images in Fig 5c are representative of the data shown in Fig 5b but this doesn't make sense as each mutant strain has no TssK foci in any of the cells in Fig 5c but in the graph in Fig 5b there are still some foci detected in all mutant backgrounds.

As discussed above in response to reviewer 1, we have replaced the anti-TssK data in the manuscript with new data from analysis performed with strains expressing TssB-sfGFP. We have now added representative micrographs for Figure 5A. Reviewer 3 is correct in that showing close zooms of a few cells resulted in the nuance of occasional foci in the mutant backgrounds to be lost. Our micrographs in Figure 5C and 5E now show larger fields with more cells, with only occasional foci detected in the mutant strains.

The authors should also provide a citation(s) for the statement in line 273 “...consistent with previously reported frequencies in other bacteria.”

As suggested, we have now cited multiple prior studies which examined the frequency of T6SS foci (of varying types) in different species of bacteria, including Brunet et al. *PLoS Genet.* 2015 = DOI:10.1371/journal.pgen.1005545 (at least 2-3 foci/cell), Logger et al. *JMB* 2016 = DOI : 10.1016/j.jmb.2016.08.032 (1 foci/cell), Cherrak et al. *Nature Micro.* 2018 = DOI: 10.1038/s41564-018-0260-1 (mean 1.5 foci/cell), Cherrak et al. *mBio.* 2021 = DOI: 10.1128/mBio.01348-21 (at least 2-3 foci/cell).

Lines 277-279 and Figure 5d-e: The same is true in Fig 5d-e as for the above point in Figs 5b-c: the apparently representative images in Fig 5e do not line up with the number of foci/cells reported in 5d i.e. there are still some foci in the low expression condition but no foci for this same condition in Fig 5e. As above, this should be addressed by including larger fields of view from the microscopy image that has more than just 2 cells per field.

This comment has been addressed above in our newly presented TssB-sfGFP data, which include larger fields of view and higher quality images.

Minor comments:

Line 161: the authors claim at this point to have identified “five membrane associated proteins”. At this point in the manuscript the data only demonstrates that these proteins are essential for T6SS function and that they are likely membrane associated from in silico analyses but not yet demonstrated to be membrane associated. I would suggest to revise to say “five putative membrane associated proteins”.

We agree with reviewer #3. Lines 161-163: We have rephrased such as “In conclusion, we have identified five putative membrane associated proteins, conserved in Bacteroidales, that are essential to the function of the *B. fragilis* T6SSⁱⁱⁱ”.

Figure 2a: It would be helpful to define what the abbreviations are for each column title in the figure legend to help the reader understand the information in the table.

We have modified the legend of Figure 2 to better define the abbreviations (lines 898-910).

Figure 3 and lines 861-863: the authors are proposing elsewhere in the manuscript that the whole complex is anchored to the outer membrane by putative lipoprotein TssR – in this figure there is a pink arm extending horizontally off TssR but it might make more sense if this arm is actual vertical and contacting the outer membrane in the figure? Also the figure legend for this says that “The full arrows represent copurification observed in full length proteins, the dashed arrow represent copurification of the periplasmic truncations of the proteins.” However, the lines in the actual figure are black with arrow heads or grey with blunt ends – this should be corrected in the figure/legend so both match.

We have modified Figure 3 and the corresponding legend following reviewer #3’s advice.

Figures 5c and 5e: the left-hand image in both cases is the composite of the TssK fluorescent image and the phase contrast image – it is really hard to actually see the TssK fluorescent foci in these composite images over the background fluorescence in the whole cell, compared to the much clearer fluorescent foci in the middle images. Why is this? Have the authors altered the brightness and contrast or maximum levels of this fluorescent channel differently in the left-hand composite image compared to what is presented in the middle image? I would suggest correcting this so the foci are clear in the composite image, or alternatively remove the composite images as it is quite confusing.

We have overhauled this section of the manuscript using new strains in order to improve image quality and interpretability (see responses to reviewers above). We have added analysis with TssB-sfGFP (sheath protein), which we feel has stronger signal and less background, leading to more robust data. Importantly, we have arrived at the same conclusions we had made using the original anti-TssK data. In general, however, we do not differentially alter signal brightness between micrographs - each image has been set to the same signal threshold for display.

Line 442: “...found in protein...” should be “...found in proteins...”

We have modified the text accordingly.

Reviewer #1 (Remarks to the Author):

In the revised version of the manuscript, Bongiovanni and colleagues have provided additional support for their fluorescent microscopy observations and answered most of the concerns of the reviewers. I agree with the other reviewers, that at this point, authors have done probably almost everything that is possible to support their hypothesis about the composition of the T6SSiii membrane complex. The manuscript is well written and this extensive analysis will be appreciated in the field.

Reviewer #2 (Remarks to the Author):

The revised manuscript addresses my concerns from the first review. The inclusion of the TssB-GFP data instead of the Tssk immunostaining is much clearer.

One small suggestions though for the presentation of the newly added heatmap data would be to include cell outline or some sort of labels for the cell poles or the long/short axis of the cell.

Reviewer #3 (Remarks to the Author):

See the attached document.

Reviewer #3 Attachment on the following page

Bongiovanni et al revised manuscript

I thank Bongiovanni et al. for taking the time to address my comments and queries. Overall the authors have almost addressed all of my comments, I just have a couple of issues that I think need to be addressed as outlined below. I have included my previous comments (in black text), the authors' response (in blue text), and then my response to the revised version of the manuscript (in purple text).

Major comments:

Lines 169-170: Did the authors confirm at any point that the STREP and HIS tags on TssN and TssO at the native location did not impact T6SS function? The authors should use their T6SS competition assay and the Hcp secretion ELISA assay to confirm that the tagged versions of these proteins have wildtype T6SS functionality.

The reviewer raises a good point. To address this, we have performed competition experiments in which we assess competitive index of single- and doubly-tagged strains in comparison to the wildtype. We find that each single tag exhibits wildtype T6SS function. Although the double-tagged strain is slightly diminished in function, it still retains ~100fold higher functionality compared to a T6SS-inactive sheath-mutant control strain. We have included these data in the supplemental figures and add the following sentence (line 169): "without altering the functionality of the T6SS (Extended Data Fig. 1b)."

In Extended Data Fig 1b of the revised manuscript which has competition data to show that the STREP and HIS tags on TssN and TssO at the native locus do not impact upon T6SS functionality – there are no labels on the x-axis or in the figure legend to denote what each bar is. Therefore, it is not clear to me how to interpret the data and thus if this data is showing what they claim in their response to reviewers comment above.

2. Lines 274-282 and Figure 5a-c: The authors should show representative fluorescent images of the TssK foci that were used to determine the number of foci/total cells in Figure 5a as has been done for Fig 5b with the images in Fig 5c. In both cases it would be better to include a larger region of the field of view for this than just one or two cells – the authors claim that the images in Fig 5c are representative of the data shown in Fig 5b but this doesn't make sense as each mutant strain has no TssK foci in any of the cells in Fig 5c but in the graph in Fig 5b there are still some foci detected in all mutant backgrounds.

As discussed above in response to reviewer 1, we have replaced the anti-TssK data in the manuscript with new data from analysis performed with strains expressing TssB- sfGFP. We have now added representative micrographs for Figure 5A. Reviewer 3 is correct in that showing close zooms of a few cells resulted in the nuance of occasional foci in the mutant backgrounds to be lost. Our micrographs in Figure 5C and 5E now show larger fields with more cells, with only occasional foci detected in the mutant strains.

I appreciate the effort taken by the authors to improve their fluorescence microscopy images and data. The new TssB-sfGFP foci are certainly a lot clearer than the previous microscopy. It is also an improvement that the authors have included larger fields of view for the representative images to support the data. However, I do not think that the images chosen are necessarily representative of the data presented. Specifically, in Fig 5a the average number of cells with TssB-sfGFP foci in the graph is approx. 7% and ranging from approx. 2-25% however the number of cells with foci in the representative microscopy image has 14/19 cells (77%) with foci – how is this a representative image reflective of the quantification data shown in Fig 5a? Additionally, in Fig 5c the top image in the panel is TssB-sfGFP and it appears that a large amount of cells in this field have foci (approx.. 17/35 = 40%) – again

shouldn't this strain have around 5% of cells with foci (as is presented in the graph in Fig 5a)? I would suggest that the authors select images that are a better representation of the quantified data.

3. Lines 277-279 and Figure 5d-e: The same is true in Fig 5d-e as for the above point in Figs 5b-c: the apparently representative images in Fig 5e do not line up with the number of foci/cells reported in 5d i.e. there are still some foci in the low expression condition but no foci for this same condition in Fig 5e. As above, this should be addressed by including larger fields of view from the microscopy image that has more than just 2 cells per field.

This comment has been addressed above in our newly presented TssB-sfGFP data, which include larger fields of view and higher quality images.

See my comment to point 2 above, I still think there is an issue with the fluorescence microscopy images not being representative images.

Minor comments:

All of my minor comments in the original submission have been addressed in this revised version.

Reviewer 2:

Reviewer #2 (Remarks to the Author):

The revised manuscript addresses my concerns from the first review. The inclusion of the TssB-GFP data instead of the Tssk immunostaining is much clearer.

One small suggestion though for the presentation of the newly added heatmap data would be to include cell outline or some sort of labels for the cell poles or the long/short axis of the cell.

Authors' answer:

We appreciate this point, however the data produced by our analysis software and summarized in these heatmaps do not include such an outline and we feel that the poles and other features of the cell are fairly apparent.

Reviewer 3:

Bongiovanni et al revised manuscript

I thank Bongiovanni et al. for taking the time to address my comments and queries. Overall the authors have almost addressed all of my comments, I just have a couple of issues that I think need to be addressed as outlined below. I have included my previous comments (in black text), the authors' response (in blue text), and then my response to the revised version of the manuscript (in purple text).

Major comments:

Lines 169-170: Did the authors confirm at any point that the STREP and HIS tags on TssN and TssO at the native location did not impact T6SS function? The authors should use their T6SS competition assay and the Hcp secretion ELISA assay to confirm that the tagged versions of these proteins have wildtype T6SS functionality.

The reviewer raises a good point. To address this, we have performed competition experiments in which we assess competitive index of single- and doubly-tagged strains in comparison to the wildtype. We find that each single tag exhibits wildtype T6SS function. Although the double-tagged strain is slightly diminished in function, it still retains ~100fold higher functionality compared to a T6SS-inactive sheath-mutant control strain. We have included these data in the supplemental figures and add the following sentence (line 169): "without altering the functionality of the T6SS (Extended Data Fig. 1b)."

In Extended Data Fig 1b of the revised manuscript which has competition data to show that the STREP and HIS tags on TssN and TssO at the native locus do not impact upon T6SS functionality – there are no labels on the x-axis or in the figure legend to denote what each bar is. Therefore, it is not clear to me how to interpret the data and thus if this data is showing what they claim in their response to reviewers comment above.

An early draft of Extended Data Fig 1 lacked the labels on the x-axis and figure legend. Our final figure includes that information.

2. Lines 274-282 and Figure 5a-c: The authors should show representative fluorescent images of the TssK foci that were used to determine the number of foci/total cells in Figure 5a as has been done for Fig 5b with the images in Fig 5c. In both cases it would be better to include a larger region of the field of view for this than just one or two cells – the authors claim that the images in Fig 5c are representative of the data shown in Fig 5b but this doesn't make sense as each mutant strain has no TssK foci in any of the cells in Fig 5c but in the graph in Fig 5b there are still some foci detected in all mutant backgrounds.

As discussed above in response to reviewer 1, we have replaced the anti-TssK data in the manuscript with new data from analysis performed with strains expressing TssB- sfGFP. We have now added representative micrographs for Figure 5A. Reviewer 3 is correct in that showing close zooms of a few cells resulted in the nuance of occasional foci in the mutant backgrounds to be lost. Our micrographs in Figure 5C and 5E now show larger fields with more cells, with only occasional foci detected in the mutant strains.

I appreciate the effort taken by the authors to improve their fluorescence microscopy images and data. The new TssB-sfGFP foci are certainly a lot clearer than the previous microscopy. It is also an improvement that the authors have included larger fields of view for the representative images to support the data. However, I do not think that the images chosen are necessarily representative of the data presented. Specifically, in Fig 5a the average number of cells with TssB-sfGFP foci in the graph is approx. 7% and ranging from approx. 2-25% however the number of cells with foci in the representative microscopy image has 14/19 cells (77%) with foci – how is this a representative image reflective of the quantification data shown in Fig 5a? Additionally, in Fig 5c the top image in the panel is TssB-sfGFP and it appears that a large amount of cells in this field have foci (approx.. $17/35 = 40\%$) – again shouldn't this strain have around 5% of cells with foci (as is presented in the graph in Fig 5a)? I would suggest that the authors select images that are a better representation of the quantified data.

We appreciate this suggestion. We have updated the wildtype micrographs to reflect the quantifications. Now, Fig 5a shows 5/27 cells with foci (18%), Fig 5c shows 5/34 (15%) for wildtype, 2/37 for *ΔtssN* (5%), 6/50 for *ΔtssO* (12%), 3/38 for *ΔtssP* (8%), 5/42 for *ΔtssQ* (11%), and 8/58 for *ΔtssR* (14%). These values are all within the ranges reflected in Fig 5b. Further, the quantification in Fig 5b has been normalized to wildtype for each corresponding experiment. We have not replaced the mutant micrographs and have kept them unchanged from the revision submission.

3. Lines 277-279 and Figure 5d-e: The same is true in Fig 5d-e as for the above point in Figs 5b-c: the apparently representative images in Fig 5e do not line up with the number of foci/cells reported in 5d i.e. there are still some foci in the low expression condition but no foci for this same condition in Fig 5e. As above, this should be addressed by including larger fields of view from the microscopy image that has more than just 2 cells per field.

This comment has been addressed above in our newly presented TssB-sfGFP data, which include larger fields of view and higher quality images.

See my comment to point 2 above, I still think there is an issue with the fluorescence microscopy images not being representative images.

The reviewer's initial comment concerned images that we presented as representative for the anti-TssK staining experiments in our original manuscript submission. We have replaced these data in our revised manuscript with TssB-GFP strains and due to the enhanced quality of these strains we can now present many more cells per representative image. Similar to the point we make in response to reviewer's point #2, we believe the TssB-sfGFP images presented are indicative of the quantifications.

Minor comments:

All of my minor comments in the original submission have been addressed in this revised version.